# Respiration modulates sleep oscillations and memory reactivation in humans

Thomas Schreiner [1] ✉, Marit Petzka[2,3], Tobias Staudigl [1] &
Bernhard P. Staresina [4,5] ✉

The beneficial effect of sleep on memory consolidation relies on the precise interplay of slow oscillations and spindles. However, whether these rhythms are orchestrated by an underlying pacemaker has remained elusive. Here, we tested the relationship between respiration, which has been shown to impact brain rhythms and cognition during wake, sleep-related oscillations and memory reactivation in humans. We re-analysed an existing dataset, where scalp electroencephalography and respiration were recorded throughout an experiment in which participants ($N = 20$) acquired associative memories before taking a nap. Our results reveal that respiration modulates the emergence of sleep oscillations. Specifically, slow oscillations, spindles as well as their interplay (i.e., slow-oscillation_spindle complexes) systematically increase towards inhalation peaks. Moreover, the strength of respiration - slow-oscillation_spindle coupling is linked to the extent of memory reactivation (i.e., classifier evidence in favour of the previously learned stimulus category) during slow-oscillation_spindles. Our results identify a clear association between respiration and memory consolidation in humans and highlight the role of brain-body interactions during sleep.

How are memories strengthened while we sleep? Current models emphasize the key role of reactivation of information encoded during prior wakefulness[1]. Through reactivation, memory representations are relayed between the hippocampus and cortical long-term stores, transforming initially labile representations into long-lasting memories[2,3]. This hippocampal–cortical communication is thought to be facilitated by the multiplexed co-occurrence of cardinal non-rapid eye movement (NREM) sleep oscillations, namely cortical slow oscillations (SOs; ~1 Hz), thalamic sleep spindles (~12–16 Hz), and hippocampal ripples (~80–120 Hz in humans)[4,5].

Recent work in humans and rodent models has corroborated the role of SO-spindle coupling during NREM sleep for both the physiological and behavioral expressions of memory consolidation. Evidence from two-photon imaging in mice suggests that the plasticity supporting role of spindles (via Ca2+ influx) is strongly amplified when

spindles concur with SO upstates[6]. Moreover, hippocampal ripples preferentially emerge when SOs and spindles are coupled[7–9], while hippocampal–cortical interactions are most prominent when preceded by SO-spindles[8]. Finally, SO-spindle events and the precision of their coupling have been shown to be instrumental for the retention of episodic memories[10–12] and to clock endogenous memory reactivation in humans[13].

In sum, the coordinated interplay of SOs and spindles has been established as a crucial cornerstone for memory consolidation. However, while spindles tend to cluster in SO upstates, the exact preferred SO phase at which spindles occur is highly variable—not only from event to event but also across individuals and development. That is, there is considerable inter-individual variability in the preferred phase of SO-spindle coupling[14]. Moreover, its precision increases from childhood to adolescence[15] and then declines again during ageing[10,16],

[1]Department of Psychology, Ludwig-Maximilians-Universität München, München, Germany. [2]Max Planck Institute for Human Development, Berlin, Germany. [3]Institute of Psychology, University of Hamburg, Hamburg, Germany. [4]Department of Experimental Psychology, University of Oxford, Oxford, UK. [5]Oxford Centre for Human Brain Activity, Wellcome Centre for Integrative Neuroimaging, Department of Psychiatry, University of Oxford, Oxford, UK. ✉e-mail: Thomas.Schreiner@psy.lmu.de; bernhard.staresina@psy.ox.ac.uk

with concomitant increases and decreases in memory performance. These findings beg the question whether there is an additional underlying pacemaker that influences SO-spindle coupling. Respiration has recently been put forward as such a scaffold for brain dynamics, as a growing number of findings demonstrate that breathing impacts cognition[17–21] and modulates brain oscillations[17,22] during wake in humans. Interestingly, the rate of respiration (i.e., breathing frequency) declines from birth to adolescence[23], while sleep-related breathing disturbances are very common in older adults[24], with the severity of symptoms accelerating with age[25]. Hence, these breathing-related changes closely parallel developmental changes in the precision of SO-spindle coupling. However, whether respiration is indeed associated sleep rhythms and ensuing consolidation processes remains unknown.

In this study, we assessed the impact of breathing on sleep rhythms and memory reactivation. Using EEG and respiratory recordings in a learning/nap paradigm, we show a clear association between respiration and the emergence of SOs, spindles as well their interplay in the form of coupled SO_spindle complexes during NREM sleep. Moreover, the strength of respiration-SO_spindle coupling is directly linked to memory reactivation during SO_spindle complexes. Our results thus identify a tight relationship between respiration and SO-spindle-mediated memory consolidation in humans.

## Results

To examine whether respiration modulates particular oscillatory signatures of human NREM sleep, we analyzed electroencephalography (EEG) and respiratory signals from 20 participants taking part in two experimental sessions. In both sessions they performed an episodic learning task, associating verbs with images of objects or scenes (counterbalanced across sessions). The learning task was followed by a nap (average sleep time: 101.63 ± 2.3 min; see Fig. 1). Each of the experimental sessions ended with a localizer task, where a new set of object and scene images was presented. This localizer served to train a linear classifier to distinguish object- vs. scene-related EEG patterns. Note that parts of the data have already been published[13], demonstrating that SO_spindles clock memory reactivation in humans. Specifics about the memory task and identification of memory reactivation during SO_spindles are thus not covered here.

### Respiration modulates NREM sleep rhythms

We hypothesized that if there is an association between respiration and sleep oscillations, this should be expressed in respiration-locked power changes in sleep EEG recordings. Hence, we detected inhalation peaks in the respiratory signal using established algorithms[17,26]. The inhalation-centered EEG data [±2 s] were then subjected to a time–frequency analysis [1–25 Hz] and contrasted against data segments randomly selected in relation to the respiration signal. Respiration-locked time–frequency representations (TFRs) exhibited increased power in the SO_spindle range around the inhalation peak, i.e., an initial low-frequency burst (comprising peaks in the SO and theta range[27–29]) followed by a fast spindle burst (12–18 Hz; $P < 0.001$, corrected for multiple comparisons across time, frequency, and electrodes; see Fig. 2a; for individual time–frequency representations of five representative participants see Supplementary Fig. 1). Applying the same analysis to source-space data suggested that these effects

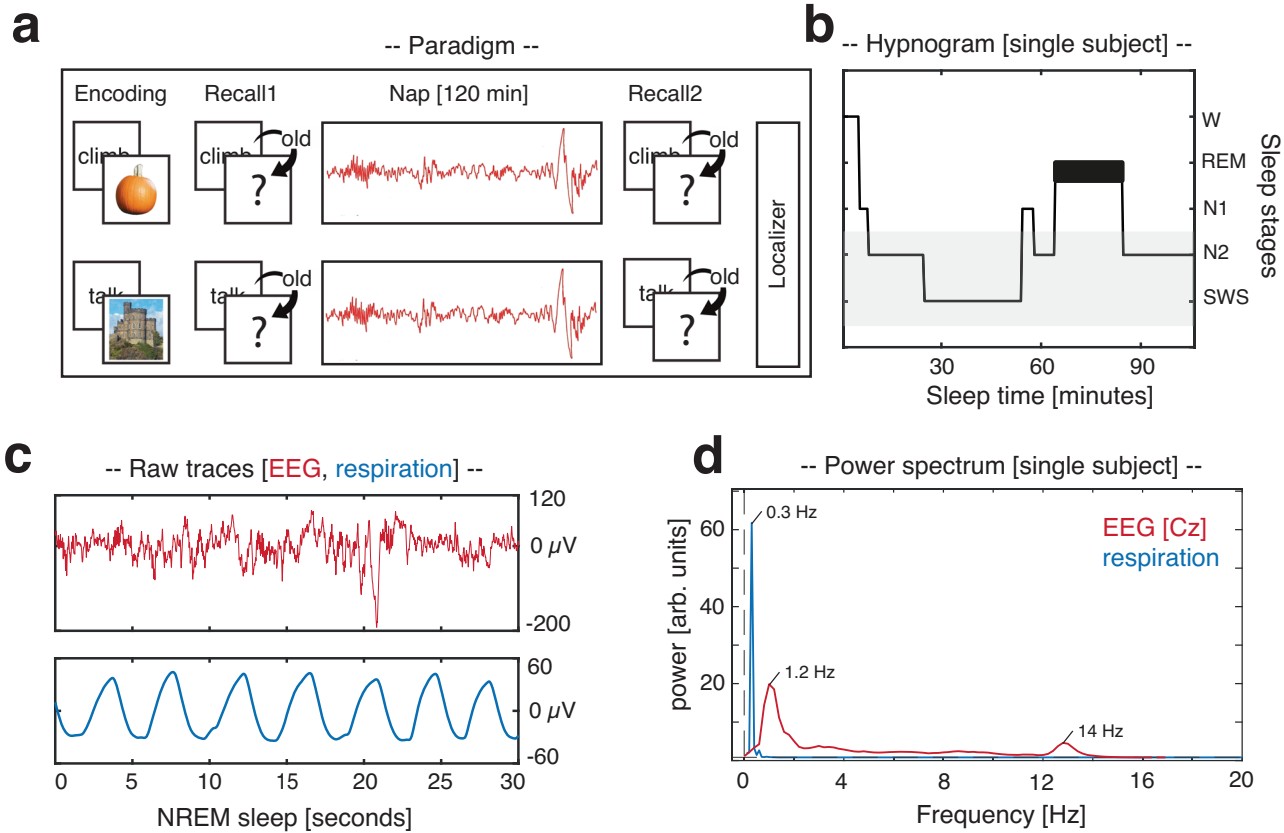

**Fig. 1 | Experimental procedure. a** During encoding, participants were presented with 120 verb-object or verb-scene pairs (counterbalanced across sessions). Memory performance was tested before and after a 120 min nap period. Each session ended with a localizer task in which participants processed a new set of object and scene images. **b** Hypnogram of a sample participant, showing time spent in different sleep stages across one nap. The gray shading indicates NREM sleep stages N2 and SWS. **c** Example of a NREM sleep segment at Cz (30 s; top row (red): EEG recording; bottom row (blue): respiration. **d** Example of a 1/f corrected power spectrum during NREM sleep at Cz for EEG (red) and respiratory recordings (blue), as obtained by Irregular Resampling Auto-Spectral Analysis (IRASA)[103]. Source data are provided as a Source Data file.

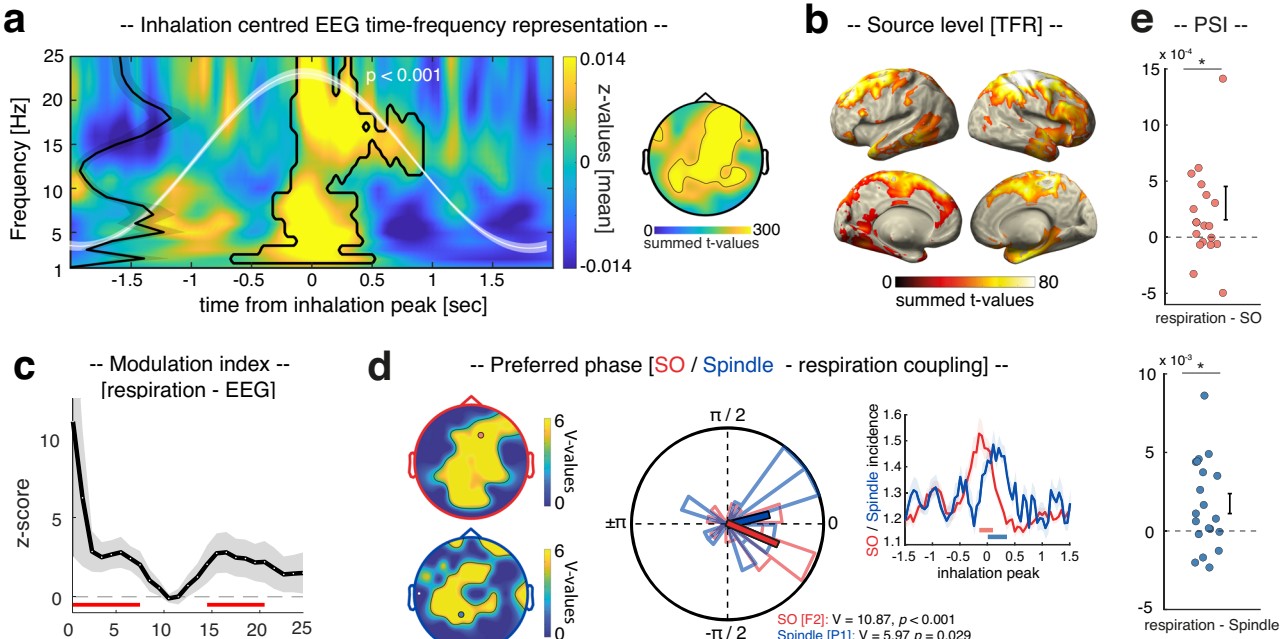

**Fig. 2 | Respiration modulates sleep rhythms. a** Time–frequency representation of NREM sleep EEG data locked to inhalation peaks, contrasted against random data segments (mean $z$ values across significant electrodes). Contours indicate significant clusters (two-sided dependent-sample $t$ test; $P < 0.001$, corrected), illustrating enhanced power in the SO_spindle range around inhalation peaks (time = 0). The black line on the left illustrates mean $z$ power, averaged across time in the cluster (±SEM across participants). The white waveform depicts the inhalation-locked respiration signal (mean ± SEM across participants). The topography illustrates the statistical results across electrodes. **b** Source data suggest that TFR results emerge from frontoparietal and right medial temporal lobe areas ($P = 0.037$, corrected). **c** The modulation index indicates that the respiration phase influences EEG amplitudes at electrode F2 (mean ± SEM across participants) with peaks in the SO (0.5 Hz) and spindle range (15.5 Hz; $z > 1.96$; the red line depicts significantly modulated frequencies; corrected). **d** SO downstates (red) and spindle onsets (blue) were non-uniformly distributed across participants in relation to the respiratory phase (two-sided V- test; SOs: $V_{mean} = 7.42 \pm 0.28$, all $P < 0.05$; Spindles: $V_{mean} = 6.84 \pm 0.24$, all $P < 0.05$; corrected using FDR[32]; contour lines encompass significant electrodes). The circular plot illustrates the preferred respiration phases for SO (red, electrode F2: mean angle = $-21.36° \pm 0.20$, mean vector length = 0.58) and spindle modulation (blue, electrode P1: mean angle = $14.06° \pm 0.23$, mean vector length = 0.44; inhalation peak = 0°). The right panel illustrates the temporal modulation of SOs (red, electrode F2) and spindles (blue, electrode P1; event percentage in relation to inhalation peaks ± 1.5 s; mean ± SEM across participants) by respiration. Solid horizontal lines indicate significant differences from event-free segments (SOs: $p = 0.004$; Spindles: $P = 0.002$; corrected). **e** The phase-slope index (PSI; mean ± SEM across participants), indicates that respiration phases predict both SO and spindle amplitudes ($t$ tests against zero, two-sided; SOs (red): $t_{1,19} = 2.17$; $P = 0.042$; Spindles (blue): $t_{1,19} = 3.1$; $P = 0.005$. Source data are provided as a Source Data file.

originated from frontoparietal areas and the right medial temporal lobe (see Fig. 2b; $P = 0.037$, corrected across time, frequency, and voxels).

We further assessed the relationship between respiration and NREM sleep EEG data using a complementary analytical approach. Specifically, we computed the modulation index (MI)[30] to estimate cross-frequency phase amplitude coupling between respiration (providing the phase) and EEG recordings at electrode F2 (providing amplitude measures between 0.5 and 24.5 Hz). Results revealed a significant modulation of EEG amplitude by the phase of respiration with local peaks in the SO (0.5 Hz) and spindle range (15.5 Hz; $z > 1.96$; corrected for multiple comparisons across frequencies, Fig. 2c).

Finally, we directly tested the association between respiration and discrete SO and spindle events. First, we identified SOs in the EEG recordings[13,31] and determined the respiratory phase during the peak of the detected SOs (downstate) across participants. A significant non-uniform circular distribution became apparent at frontal, central and parietal electrodes (V-test against 0°, $V_{mean} = 7.42 \pm 0.28$, all $P < 0.05$; corrected for multiple comparisons across electrodes using FDR correction[32], Fig. 2d), with the preferred phases of the respiration–SO modulation clustering just before the inhalation peak (i.e., 0°; mean angle of significant electrodes: $-9.4° \pm 0.22$, mean vector length = 0.51; see example circular plot and temporal modulation of SOs by respiration at electrode F2 in Fig. 2d). On an individual level we found significant nonuniform distributions in 15/20 participants. Finally, we

quantified the directional influence of respiration on SO activity at electrode F2 using the Phase Slope Index (PSI[33]). We found that respiration predicted SO activity, as evidenced by a positive PSI (mean PSI: $0.0003 \pm 0.0001$; $t$ test against zero: $t_{1,19} = 2.17$; $P = 0.042$; see Fig. 2e).

Next, spindles were identified in the EEG recordings[9,13,31] and the preferred respiration phase for spindle onsets was assessed. A significant nonuniform circular distribution was found across parietal electrodes (V-test against 0°, $V_{mean} = 6.84 \pm 0.24$, all $P < 0.05$; corrected for multiple comparisons across electrodes using FDR correction[32]), with spindle onsets clustering right after the inhalation peak (mean angle of significant electrodes: $16.95° \pm 0.23$, mean vector length = 0.52; see example circular plot and temporal modulation of spindles by respiration at electrode P1 in Fig. 2d). On an individual level we found significant nonuniform distributions in 18/20 participants. Again, we quantified the directional influence of respiration on spindle activity at electrode P1 using the PSI[33]. As with SOs, we found that respiration predicted spindle activity, as evidenced by a positive PSI (mean PSI: $0.0024 \pm 0.0007$; $t$ test against zero: $t_{1,19} = 3.1$; $P = 0.005$; see Fig. 2e).

Together, these results reveal a strong modulation of SOs and spindles by respiration, with SOs grouping at earlier respiration phases than spindles (see Supplementary Fig. 2, indicating a phase shift between SOs and spindles in relation to their modulation by respiration). An important question is whether this relationship is specific to SOs and (fast) spindles, or whether respiration is associated with any

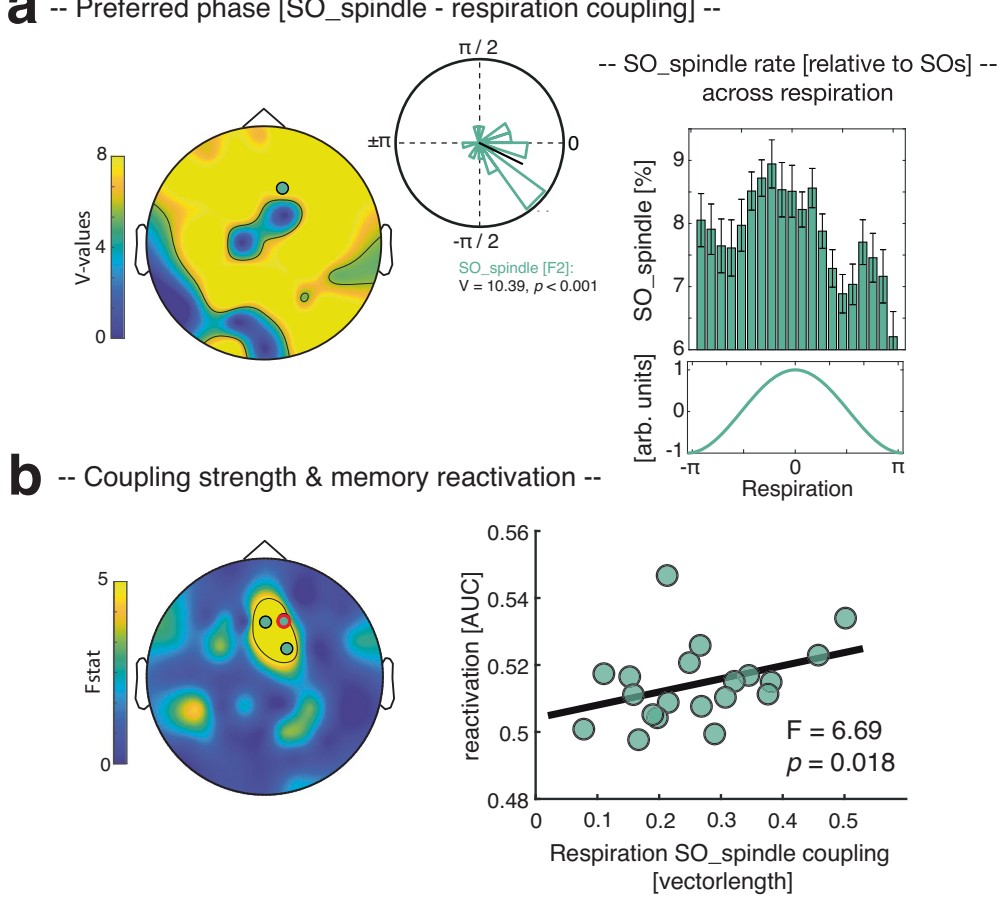

**a** -- Preferred phase [SO_spindle - respiration coupling] --

**b** -- Coupling strength & memory reactivation --

**Fig. 3 | The impact of respiration on SO_spindles and memory reactivation.**
**a** Determining the respiratory phases during the peak of the detected SO_spindles reveals a significant nonuniform circular distribution (two-sided V-test; $V_{mean} = 9.46 \pm 0.23$, all $P < 0.05$; corrected; contour lines encompass significant electrodes). The example circular plot at electrode F2 (highlighted in the topography) illustrates that the preferred phases of this respiration-SO_spindle modulation peaked right before the inhalation peak (i.e., 0°, mean angle = −24.62° ± 0.20, mean vector length = 0.57). In all participants, SO_spindle rate (relative to SOs ± SEM across participants) at F2 was non-uniformly distributed across the

respiration cycle (all $P < 0.001$; corrected, see right panel). **b** Results of a robust regression (two-sided) exhibiting a significant positive relationship between the respiration–SO_spindle coupling strength (i.e., vector length) and levels of memory reactivation (i.e., decoding accuracies as reported in ref. [13]) in a frontal cluster (all $P < 0.05$; corrected using FDR[32], contour lines encompass significant electrodes). The scatter plot illustrates the observed effects at electrode F2 ($R^2 = 0.27$, $F_{1,19} = 6.69$, $P = 0.018$; red circle on topography). Source data are provided as a Source Data file.

EEG rhythm that prevails depending on the brain state. We addressed this question by assessing whether slow spindles (9–12 Hz[34]) would be similarly modulated by respiration. Determining the respiratory phases during the onset of slow spindles across participants revealed no significant nonuniform circular distribution (all $P > 0.05$; see Supplementary Fig. 3).

**Respiration modulates SO_spindle complexes, with coupling strength being associated with memory reactivation**
As mentioned in the introduction, a growing number of empirical findings point to a particular role of SO-spindle coupling for memory consolidation. Given the clear association between respiration and SOs and spindles respectively, we next tested whether respiration would also modulate the joint presence of SOs and spindles (henceforth referred to as SO_spindle complexes). To identify SO_spindle complexes, we detected events where SO downstates were followed by sleep spindles within a time window of 1.5 s[13]. Determining the respiratory phase during the peak of the SO_spindle complexes (locked to SO downstates) revealed a significant nonuniform circular distribution encompassing frontal, central and parietal electrodes (V-test against 0°, $V_{mean} = 9.46 \pm 0.23$, $P < 0.05$; corrected for multiple

comparisons across electrodes), with SO_spindles peaking briefly before the inhalation peak (mean angle across significant electrodes: −4.5° ± 0.16, mean vector length = 0.73; see example circular plot at electrode F2 in Fig. 3a). On an individual level we found significant nonuniform distributions in 18/20 participants (for topography of preferred phases for SO_spindle modulation by respiration, see Supplementary Fig. 4). To assess whether the likelihood of SOs to group spindles is modulated by the respiration phase, we binned the number of SO_spindles relative to all SOs at electrode F2 across the respiration cycle in 20 evenly spaced bins (i.e., from -pi to pi in steps of 0.31 radians). (Non)uniformity of the resulting distribution was assessed per participant using the Kolmogorov–Smirnov test[35]. We found non-uniform distributions in all participants (all $P < 0.001$; corrected for multiple comparisons across participants using FDR correction; see Fig. 3a), indicating that the phase of respiration was associated with the emergence of coupled SO_spindle events (see Supplementary Fig. 5 for results across electrodes and Supplementary Fig. 6 for results with regards to the impact of respiration on the consistency of SO-spindle coupling).

Given that we recently demonstrated in the same dataset that SO_spindle complexes coordinate memory reactivation[13], we asked

whether there would be a functional link between the respiration–SO_spindle coupling and the reprocessing of memories during sleep. To address this question, we conducted, at each electrode, a robust regression across participants between (i) the respiration-SO_spindle coupling strength (i.e., vector length) and (ii) levels of memory reactivation (i.e., decoding accuracies averaged across the significant cluster as reported in ref. 13). As shown in Fig. 3b, we observed a significant positive relationship between the two variables at frontal electrodes ($P < 0.05$, corrected for multiple comparisons across electrodes using FDR[32]; for the relationship between respiration–SO_spindle coupling strength and behavioral levels of memory consolidation see Supplementary Fig. 7; conducting robust regressions between levels of memory reactivation and respiration–SO coupling or respiration–spindle coupling independently did not yield any significant effect, see Supplementary Fig. 8).

## Discussion

Our results unveil a putative relationship between respiration, the emergence of sleep rhythms, and memory consolidation in humans. We found that respiration modulated the emergence of SOs, spindles, and potentially their interplay in the form of SO_spindle complexes. Moreover, the strength of respiration–SO_spindle coupling was associated with the extent of memory reactivation during SO_spindles, suggesting the functional significance of respiration–brain interactions.

Many organs interact with brain rhythms. For example, the cardiovascular, gastrointestinal and respiratory systems are known to show ultradian rhythmicity, which in turn interacts with brain rhythms[36–38]. These body oscillators potentially constrain endogenous neuronal dynamics (e.g., refs. 37,39,40) and might act as common clocks, organizing neuronal activity across brain regions. Among these body rhythms, breathing has recently emerged as a potential global pacemaker for neuronal oscillations and cognition during wake, linking distinct neuronal network dynamics and facilitating information processing and transfer across distributed circuits[17,20–22,37]. However, whether respiration might play a similar role in sleep-related memory consolidation, hence be associated with the emergence of sleep oscillations in humans has remained unknown. In the current study, we tackled this question by analyzing EEG and respiratory signals which were recorded in parallel during NREM sleep (Fig. 1). Of note, using the same dataset we have recently shown that spontaneous memory reactivation in humans can be tracked during the presence of coupled SO-spindle events[13]. This allowed us to not only establish the synchronizing effect of respiration on sleep oscillations (Fig. 2), but also to potentially link it to key aspects of memory consolidation, i.e., the reactivation of prior learning material (Fig. 3).

A growing number of findings in humans demonstrate that breathing impacts cognition and memory processes during wake[18,19]. One study showed that memory was better for images that were presented during the inhalation as compared to the exhalation phase of breathing[17]. In a delayed-match-to-sample task, the respiration phase at which cue and target stimuli were presented impacted recognition performance[41]. Finally, nasal breathing during a 1-h wake rest period between study and test led to elevated memory performance as compared to mouth breathing[42], pointing to a role of nasal breathing in entraining memory-related processes, possibly via the piriform-hippocampal pathway[18].

Indeed, coordinating influences of breathing on brain rhythms have been demonstrated in several brain areas linked to memory processes, including not only the hippocampus[17,43–45], but also thalamus[46] and prefrontal cortex[22,47]. Notably, these three regions also represent key structures for sleep-related memory consolidation, giving rise to ripples, sleep spindles, and SOs, respectively[48–50]. In fact, recent work in rodents has demonstrated that respiration modulates the coordination of hippocampal sharp-wave ripples and cortical down/upstate transitions during NREM sleep[46], while theta and gamma and their interplay during REM sleep are likewise impacted by breathing[51,52]. Hence, converging evidence across species points to a role of respiration in online and offline memory processing, with the latter presumably afforded by its coordinating effect on sleep rhythms supporting memory consolidation.

The precise coupling between NREM sleep oscillations (i.e., SOs, spindles, and ripples) has long been assumed to play a key part in the memory function of sleep, as it is thought to enable the information transfer between the hippocampus and cortical networks[1,5,53–56]. Indeed, work in humans has revealed that the precision of SO-spindle coupling, i.e., the exact timing of spindle maxima with respect to the SO upstate, is tightly associated with the retention of declarative learning material[10–12]. Moreover, work in rodents has demonstrated that precise SO-spindle coordination is key for sustaining the reactivation of neural ensembles[57], while the full oscillatory hierarchy (i.e., SOs, spindles, and ripples) has been shown to be necessary for effective consolidation[7,58]. Our result that respiration is related to the emergence of SOs, spindles and potentially SO_spindle coupling (and putatively the effect of the latter on memory reactivation) suggests breathing as a potential oscillatory scaffold for memory consolidation in humans. Due to the restrictions of scalp EEG, our current data remain agnostic as to whether hippocampal ripples in humans are likewise influenced by respiration, as shown in rodents[43,46]. Hence, future work will need to employ simultaneous recordings from the hippocampus to directly test whether respiration is indeed associated with the full hierarchy of NREM sleep oscillations in humans.

Is respiration associated with any EEG rhythm that prevails during a given brain state? Our result that slow spindles, which are dominant over frontal areas and emerge preferentially at the transition into the SO downstate[34], were not robustly impacted by respiration (Supplementary Fig. 3) points to a specific effect on SOs and fast spindles.

How might breathing exert its impact on brain activity? When we inhale, the incoming airflow stimulates mechanoreceptors of the olfactory sensory neurons[59]. These in turn produce breathing-locked oscillations, which are conveyed to the olfactory bulb and further to the olfactory cortex[60]. The olfactory cortex, however, is not the last terminal of breathing-entrained rhythms. The impact of breathing on neuronal activity has been identified in rodent models and humans in various brain areas[20–22,43,44,46,47,61]. This anatomical circuitry is in line with recent findings indicating a privileged role of nose breathing as compared to mouth breathing in synchronizing neuronal oscillations and affecting memory processes[17,42]. However, the brainstem houses respiratory rhythm generators, which might likewise account for the breathing-related entrainment of neuronal oscillations[46,62] (irrespective of nose or mouth breathing). Specifically, the phases of the respiratory cycle, comprising inhalation and exhalation, are governed by brainstem circuits[62]. The brainstem, as major control hub of the autonomic nervous system, likewise impacts the activity of other vital functions such as the heart rate or gastric functions[63,64]. Cardiac activity and respiration are intimately linked[65], while cardiac rhythms and in particular heart rate variability have been shown not only to covary with different sleep stages but also with SO and sleep spindle activity during NREM sleep[66]. Hence, the extent to which sensory (i.e., olfactory bulb route) or non-sensory (i.e., brainstem route) breathing-locked inputs modulate neuronal activity and whether they innervate the same target regions in the brain remains unclear. Controlling the breathing route during sleep would permit drawing more causal inferences whether nose breathing is indeed key for clocking neuronal activity during sleep.

On that note, it deserves mention that our results are correlational in nature. Hence, future studies, directly manipulating breathing behavior, will be needed to provide causal evidence for the role of breathing in modulating sleep-related oscillations. For example, it has been shown that presenting odors during NREM sleep is capable of

modifying respiration during sleep, by decreasing inhalation and increasing exhalation volume for up to 6 breaths[67]. One hypothesis would be that such interference (i.e., shallower inhalation due to odor presentation) might mitigate the modulation of sleep-related oscillations during the initial presence of odors. Another line of research potentially providing causal evidence could stem from patients suffering from obstructive sleep apnea (OSA). OSA is characterized by repetitive upper airway collapse resulting in intermittent hypoxia[68]. Strikingly, in OSA patients' sleep seems to have lost its beneficial effect on memory[69]. While it has been shown that abnormal sleep spindle properties are associated with OSA[70], it remains unclear whether OSA also impacts the emergence of SOs and SO_spindle complexes. This would corroborate the notion that pathological, arrhythmic breathing affects the coordination of sleep oscillations. Importantly, showing that restoring normal breathing behavior (e.g., via Continuous Positive Airway Pressure (CPAP) treatment) also restores the relationship between respiration and sleep oscillations would lend some causal evidence for the role of breathing on the coordination of sleep oscillations.

It has to be noted that we did not explicitly screen for sleep-disordered breathing (SDB) conditions in our participants, which might have considerable impacted the results of our work. However, SDB is well known to cause sleep fragmentation (e.g., ref. [71]). All our participants exhibited healthy sleep and no signs of sleep fragmentation (i.e., elevated number of awakenings) as assessed with sleep scoring. Moreover, we carefully inspected the respiratory data during recording and offline pre-processing and found no signs of breathing cessation. Finally, sleep oscillations and specifically sleep spindle properties (i.e., frequency and topography) have been shown to be altered by SDB (e.g., ref. [70]). Spindle characteristics were generally within the expected range, both in terms of frequency and topography. That said, the fact that pulse oximetry or plethysmography were not collected during measurements, while no trained clinician evaluated the data, constitutes a limitation of the current study, as we cannot exclude any influence of SDB on our results with certainty.

Another potential caveat is that 17 of our 20 participants were female. It is well known that sex differences affect sleep parameters, sleep-related oscillations and in consequence memory consolidation[72], which might limit the generalizability of our results. Even though removing the data from male participants did not change the main outcomes of our study (see Supplementary Fig. 9), future work will need to address potential gender-related differences in this context.

In recent years, several experimental procedures have been employed in an effort to non-invasively bolster sleep and strengthen overnight memory retention[73]. Entraining SOs by applying auditory clicks during SO upstates (closed-loop protocol) or gentle rocking stimulation has been shown to augment SO activity, elicit coupled sleep spindles and support memory retention[31,74,75] but see refs. [76,77]. In parallel, targeted memory reactivation (TMR) studies have established that presenting auditory reminder cues during NREM sleep strengthens the consolidation of memories[78–82]. Intriguingly, combining TMR with closed-loop procedures, i.e., placing reminder cues towards SO upstates, has been shown to be most efficient in modulating memory retention[83–85]. The association between respiration and sleep oscillations might thus be harnessed to further improve such efforts of ameliorating sleep and memory consolidation. Specifically, as inhalation tended to precede the emergence of SO-spindles, TMR procedures could capitalize on the enhanced accessibility/signal-to-noise ratio of respiration to optimize experimental protocols for memory reactivation.

## Methods

The current analyses are based on ref. 13 and detailed information about participants, stimuli, task, data acquisition, and behavioral results can be found in the original article. In brief, twenty healthy participants (mean age: 20.75 ± 0.35; 17 female) took part in the experiment (for results comprising only female participants see Supplementary Fig. 9). The data of five additional participants had to be excluded due to insufficient sleep (less than 30 min sleep during one of the sessions). No statistical method was used to predetermine sample size. Pre-study screening questionnaires (including the Pittsburgh Sleep Quality Index (PSQI[86]), the morningness–eveningness questionnaire[87], and a self-developed questionnaire inquiring general health status and the use of stimulants indicated that participants did not take any medication at the time of the experimental session and did not suffer from any neurological or psychiatric disorders. All participants reported good overall sleep quality. The study was approved by the University of Birmingham Research Ethics Committee and written informed consent was obtained from participants. Participants received financial compensation or course credit for their participation in the study.

### Experimental overview

The experiment consisted of two experimental sessions, separated by at least 1 week (mean = 8.5 ± 0.85 days). The order of the two sessions was counterbalanced across participants. The investigators were not blinded to the order of session allocation. On experimental days participants arrived at the sleep laboratory at 11 a.m. The experimental sessions started with the set-up for polysomnographic recordings during which electrodes for electroencephalographic (EEG), electromyographic (EMG), and electrocardiographic (ECG) recordings were applied. In addition, a thermistor airflow sensor was attached to record breathing. Before the experimental sessions, participants were habituated to the environment by spending an adaptation nap in the sleep laboratory. At around 12 p.m. the experiment started with a modified version of the psychomotor vigilance task (PVT[88]), followed by the memory task, where participants learned to associate 120 verbs and images (comprising objects or scenes[89], depending on the experimental session). The sleep period began at ~1 p.m. and participants were given 120 min to nap (mean total sleep time: 101.63 ± 2.23 min). Afterward, the vigilance of all participants was assessed using the PVT and memory performance was tested again. At the end of each session a localizer task was conducted. For the recording of behavioral responses and the presentation of all experimental tasks, Psychophysics Toolbox Version 3[90] and MATLAB 2018b (MathWorks, Natick, USA) were used.

### EEG and respiration

A Brain Products 64-channel EEG system was used to record electroencephalography (EEG) throughout the experiment. Impedances were kept below 10 kΩ. EEG signals were referenced online to electrode FCz and sampled at a rate of 1000 Hz. Furthermore, EMG and the ECG was recorded for polysomnography. Respiration was recorded using an Embla thermistor airflow sensor. Sleep architecture was determined offline according to standard criteria by two independent raters[91].

### Data analysis

All data were analyzed using Matlab (2018b; Mathworks). EEG data were preprocessed using the FieldTrip toolbox for EEG/MEG analysis[92] (v.09/01/2020). All data were downsampled to 200 Hz. Noisy EEG channels were identified by visual inspection, discarded, and interpolated, using a weighted average of the neighboring channels. Following standard procedures, all sleep data were re-referenced against linked mastoids. Subsequently, the sleep data were segmented into 4 s epochs and time-locked to inhalation peaks [−2 to +2 s] as derived from the respiratory signal. We used electrode F2 as representative electrode for all SO and SO_spindle-related analyses and electrode P1 for spindles. In Supplementary Fig. 10, we show the average of all significant electrodes for phase-related analyses and in Supplementary Fig. 11 we show the electrodes exhibiting the strongest phase-modulation related effects.

## Event detection

SOs, sleep spindles and inhalation peaks were identified for each participant, based on established detection algorithms[13,17,31,93,94]. SOs were detected as follows: Data were filtered between 0.3 and 1.25 Hz (two-pass FIR band-pass filter, order = three cycles of the low-frequency cutoff). Only movement-free data (as determined during sleep scoring) from NREM sleep stages 2 and 3 were considered. All zero-crossings were determined in the filtered signal and event duration was determined for SO candidates (that is, downstates followed by upstates) as time between two successive positive- to-negative zero-crossings. Events that met the SO duration criterion (minimum of 0.8 and maximum of 2 s) entered the analysis.

For spindle detection, data were filtered between 12 and 18 Hz (two-pass FIR band-pass filter, order = three cycles of the low-frequency cutoff), and again only artifact-free data from NREM sleep stages 2 and 3 were used for event detection. Note that we used a slightly wider frequency range (12–18 Hz as opposed to the more traditional 12–16 Hz) for consistency with the detection settings previously applied to the same data[13] and to accommodate the fact that fast spindles tend to exceed 16 Hz in humans[13,95,96]. The root mean square (RMS) signal was calculated for the filtered signal using a moving average of 200 ms, and a spindle amplitude criterion was defined as the 75% percentile of RMS values. Whenever the signal exceeded this threshold for more than 0.5 s but less than 3 s (duration criterion), a spindle event was detected. To isolate SO-spindle complexes, we determined for all SOs whether a spindle was detected following the SO (SO downstate + 1.5 s). For statistical comparisons, we also extracted 4-s- long intervals during NREM sleep which did not exhibit any SO, spindle or SO-spindle event, respectively. Event-free segments were only drawn from time window starting 5 min before and ending 5 min after the corresponding oscillatory event. Inhalation peaks were detected using BreathMetrics[26].

## Time–frequency analysis

Time–frequency analysis of inhalation peak-centered EEG segments [−4 to 4 s] was performed using FieldTrip[92]. The longer time segments were chosen to allow for resolving low-frequency activity within the time windows of interest [−2 to 2 s] and avoid edge artifacts. Frequency decomposition of the data was achieved using Fourier analysis based on sliding time windows (moving forward in 50 ms increments). The window length was set to five cycles of a given frequency (frequency range: 1–25 Hz in 1 Hz steps). The windowed data segments were multiplied with a Hanning taper before Fourier analysis. Afterward, power values were z-scored across time [−2 to 2 s].

## Respiration–sleep oscillation coupling

For the analysis of the coupling between respiration and the sleep graph elements (SOs, spindles and SO-spindles), we determined in each participant the respiratory peak frequency (mean = $0.25 \pm 0.005$ Hz) and filtered the respiratory data (locked to inhalation peaks) around the peak frequency ($\pm 0.05$ Hz, two-pass Butterworth band-pass filter, order = three cycles of the low-frequency cutoff). Then a Hilbert transform was applied, and the instantaneous phase angle was extracted (for outcomes based on extracting respiratory phase using a double interpolation[21] approach see Supplementary Fig. 12). Next, we isolated the respiratory phase angle at the time of SO downstates (in case of respiration–SO coupling and respiration–SO_spindle coupling) and spindle onsets (in case of respiration–spindle coupling). Each participant's preferred respiratory phase at SO downstates/spindle onsets was obtained by taking the circular mean of all individual events' preferred phases.

## Modulation Index

Phase amplitude coupling was assessed with the Modulation Index (MI)[30]. To estimate instantaneous phase of the respiration signal, we filtered the continuous respiratory data around $0.25 \pm 0.05$ Hz two-pass Butterworth band-pass filter). We then extracted instantaneous amplitude data across frequencies between 0.5 and 24.5 Hz at electrode F2 in steps of 1 Hz. To this end, a two-pass FIR filter (order = three times the lower frequency bound) was used to create 20 equally spaced frequency bins, with center frequencies ranging from 0.5 to 24.5 Hz, and with fixed-frequency bandwidths of 1 Hz. The envelope of the Hilbert-transformed band-pass filtered data was then used as amplitude estimate.

To compute the MI (for a given frequency pair), we divided the phase signal into 20 bins, and then, for each bin, computed the mean amplitude for that bin. This yields a distribution of amplitude as a function of phase. The MI is defined as the Kullback–Leibler distance between that distribution and the uniform distribution (over the same number of bins). To assess the statistical significance of the MI values, we randomly shifted the phase time series, and computed the MI using the shifted signal[97]. We repeated this procedure 200 times, resulting in a MI-level reference distribution. The mean and standard deviation across the reference distribution was then used to z score the MI of the empirical data. Z values were than transformed into P values, with a significance threshold of z values greater or smaller than +/− 1.96.

## Phase-slope index

We assessed whether respiration would influence activity in the SO and sleep spindle range or vice versa using the PSI[33]. The cross-frequency PSI was calculated between the respiration signal (filtered around the peak frequency ± 0.05 Hz) and electrode F2 in the case of SOs (filtered around 0.3–1.25 Hz) and electrode P1 in case of spindles (filtered around 12 and 18 Hz). In this context, positive values indicate respiration driving SO/sleep spindle activity, while negative values indicate sleep spindles/SOs driving respiration. The obtained data distributions were tested against zero, using paired samples t tests.

## Source analysis

To estimate the sources of the obtained effects in the scalp EEG study, we applied a LCMV beamforming method, as implemented in FieldTrip[92]. A spatial filter for each specified location (each grid point; 10mm3 grid) was computed based on the cross-spectral density. Electrode locations for the 64-channel EEG system were co-registered to the surface of a standard MRI template in MNI (Montreal Neurological Institute) space using the nasion and the left and right preauricular as fiducial landmarks. A standard leadfield was computed using the standard boundary element model[98]. The forward model was created using a common dipole grid (10-mm³ grid) of the gray matter volume (derived from the anatomical automatic labeling atlas[99] in MNI space, warped onto standard MRI template, leading to 1457 virtual sensors. Frequency decomposition of the data in source space was achieved using Fourier analysis based on sliding time windows (moving forward in 100 ms increments). The window length was set to five cycles of a given frequency (frequency range: 1–25 Hz in 2 Hz steps). The windowed data segments were multiplied with a Hanning taper before Fourier analysis. Afterward, power values were z-scored across time [−2 to 2 s].

## Peri-event time histograms

To assess the temporal relationship between respiration and sleep oscillations (SOs and spindles, Fig. 2d), we created peri-event time histograms (bin size = 50 ms) where inhalation peaks served as seed (time = 0), while the targets (SO downstates and spindle onsets, respectively) are depicted relative to the seed. The resulting histograms were normalized by dividing the number of detected events per bin by the total number of detected sleep events in the time window of interest (i.e., inhalation peak ± 1.5 s). The resulting values were multiplied by 100. This was done per participant and electrode site to account for the overall rate of events (SOs or spindles, respectively) at a given site.

## SO_spindle rate across the respiration cycle

To assess whether the likelihood of SOs to group spindles is modulated by the respiration phase (Fig. 3a), we determined in each participant the respiratory peak frequency (mean = $0.25 \pm 0.005$ Hz) and filtered the respiratory data (locked to inhalation peaks) around the peak frequency ($\pm 0.05$ Hz, two-pass Butterworth band-pass filter, order = three cycles of the low-frequency cutoff). Then a Hilbert transform was applied, and the instantaneous phase angle was extracted. Next, we binned the number of detected SO_spindles at electrode F2 across the respiration cycle in 20 evenly spaced bins (i.e., from -pi to pi in steps of 0.31 radians). The same procedure was conducted with regard to all detected SOs. Finally, the resulting SO_spindle distribution was normalized per participant by dividing the number of detected SO_spindles per bin by the number of detected SO per bin. The resulting values were multiplied by 100. This was done per participant and electrode site to account for the overall rate of SOs at a given site.

## Multivariate analysis (brief description, for details, see ref. 13)

Multivariate classification of single-trial EEG data was performed using MVPA-Light, a MATLAB-based toolbox for multivariate pattern analysis[100]. For all multivariate analyses, a LDA was used as a classifier[100]. Prior to classification, all data were re-referenced using a common average reference (CAR).

To investigate differential evidence for object vs. scene representations as a function of prior learning during SO-spindle complexes, we used the temporal generalization method[101]. Prior to decoding, a baseline correction was applied based on the whole trial ([−0.5 to 3 s] for localizer segments; [−1.5 to 1.5 s] for SO-spindle segments). Next, localizer and sleep data were z-scored across trials and collapsed across sessions. PCA was applied to the pooled wake-sleep data and the first 30 principal components were retained. Localizer and sleep data were smoothed using a running average window of 150 ms. A classifier was then trained for every time point in the localizer data and applied on every time point during SO-spindle complexes. No cross-validation was required since localizer and sleep datasets were independent. As metric, we used the area under the curve. For statistical evaluation, surrogate decoding performance was calculated by shuffling the training labels (stemming from the localizer task) 250 times. Again, the resulting performance values were averaged, providing baseline values for each participant under the null hypothesis of label exchangeability.

## Statistics

Before entering statistical assessment, data were collapsed across both sessions per participant, resulting in 20 datasets. Unless stated otherwise, we used non-parametric cluster-based permutation tests, which do not rely on the assumption of a specific underlying distribution, to correct for multiple comparisons as implemented in FieldTrip[92]. A dependent-sample $t$ test was used at the sample level and values were thresholded at $P = 0.05$ (1000 randomizations). The sum of all $t$ values in clusters served as cluster statistic and Monte Carlo simulations were used to calculate the cluster $P$ value (alpha = 0.05, two-tailed) under the permutation distribution. The input data were either time × frequency x electrode/voxel values (Fig. 2a, b) or occurrence probabilities across time (e.g., Fig. 2d), which were either tested against randomly centered data segments (Fig. 2a, b) or against data stemming from event-free events. For circular statistics (e.g., 2d and 3a), the phase distributions across participants were tested against uniformity with a specified mean direction (i.e., 0°, corresponding to the inhalation peak) using the V-test (CircStat toolbox[102], v1). The unimodality of the phase distribution in relation to each stimulus category (SOs, spindles and SO-spindles) was validated using Watson's test against a von Mises distribution (all $P > 0.1$). To assess (non)uniformity of SO_spindle rate across the respiration cycle (Fig. 3a) the non-parametric Kolmogorov–Smirnov test was applied within each participant[35]. To assess the link between (i) the respiration–SO_spindle coupling strength (i.e., vector length) and (ii) levels of memory reactivation a robust regression, which reduces the impact of violations of the distribution assumption and heterogeneity in variance, was performed at each electrode to minimize the influence of outliers. The false discovery rate (FDR) was used to correct for multiple comparisons across electrodes (in case of circular statistic, e.g., 2 d, 3 a and 3b) or participants (Fig. 3a; SO_spindle rate across respiration)[32].

## Reporting summary

Further information on research design is available in the Nature Portfolio Reporting Summary linked to this article.

## Data availability

This study comprises the re-analysis of an existing dataset[13]. All processed data supporting the findings of this study are publicly available at the Open Science Framework (https://doi.org/10.17605/OSF.IO/D6YHB)[104]. While publicly sharing the raw data is prohibited due to ethics protocols, data may be shared upon request. To obtain the data, please contact the corresponding author, Thomas Schreiner (Thomas.Schreiner@psy.lmu.de). Source data are provided with this paper.

## Code availability

All custom codes to reproduce the central findings of this study are available at the Open Science Framework (https://doi.org/10.17605/OSF.IO/D6YHB)[104].

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

## Acknowledgements
T.Sc. is supported by the Emmy Noether program of the German Research Foundation (492835154). T.St. research is funded by an ERC Starting Grant (802681). B.P.S. is supported by an ERC Consolidator grant (101001121).

## Author contributions
T.Sc. and B.P.S. conceived the study and designed the experiment. T.Sc. conducted the experiment. T.Sc., M.P., T.St., and B.P.S. analyzed the data and wrote the paper.

## Competing interests
The authors declare no competing interests.
