## [Peer Review File · Nature Communications]

Respiration shapes sleep oscillations and memory reactivation in humansREVIEWER COMMENTS

Reviewer #1 (Remarks to the Author):

Schreiner and colleagues present findings from a re-analysed human EEG study with concomitant respiration recordings during nREM sleep. In addition to respiration phase-locked clustering of sleep-related EEG signatures (slow oscillations, spindles), the authors report a link between the strength of this coupling and the extent to which memory content was subsequently reactivated during sleep. The underlying research question is of great interest for a broad audience across disciplines and addresses a critical gap in the literature. The study is well done and the reported analyses are concise, yet highly competently conducted and therefore convincing. That being said, I felt that the manuscript would benefit from additional details here and there. Moreover, some methodological details presumably deserve explanation and/or some kind of control analysis. Hoping they are of help in revising the manuscript, I list detailed suggestions and comments below.

Daniel S. Kluger

Major points

1. Respiration methodology

Some aspects of how the respiratory data were preprocessed and analysed are not uncontested in the field. I felt that three instances in particular deserve discussion and/or a control analysis to rule out contamination of the results by certain choices made in the analysis pipeline:

a) The authors employed Hilbert transform to extract the continuous phase of the respiration time series, presumably a built-in function from the BreathMetrics toolbox (although I could be wrong here). Due to inherent characteristics of human respiration, the Hilbert transform is known not to be ideal for this application, and several alternatives have been developed (single and dual interpolation approaches, prophase, and others). While there is no clear consensus regarding best practice, the many data sets from different labs/modalities on which I have seen Hilbert transform fail (and which I am happy to share with the authors, if desired) raise some scepticism about the potential influence of the phase extraction method. It would be great if the authors could report whether and how they made sure that respiration phase was reliably extracted. As a control, I suggest to repeat a single analysis (e.g., the single-sensor analysis of preferred phase shown in Fig. 2d) using a different phase extraction approach, for example by linearly interpolating between peaks and troughs as well as between troughs and peaks (dual interpolation) and compare the resulting polar histograms.

b) For the computation of the modulation index, what were the constraints used for the 'random shifts', i.e. were (sub-)harmonics of the individual breathing frequency excluded or was the width of possible shifts otherwise restricted to a certain range? As a side note, was it really the case that both time series were shifted or was one kept constant and the other was shifted?

c) The statistics reported for preferred phase are based on the circular means of polar histograms and corresponding tests (V-test against zero). Circular means work well when (and only when) the underlying distribution has a rather clear unimodal characteristic (especially when tested against a known direction with a V-test); for multimodal distributions, the circular mean may yield artificial results (see e.g. Landler et al., Sci Rep 2021). It is rather hard to guess from Fig. 2d and 3a, but it would be very convincing if the authors could report a quick control, e.g. i) using the circular median instead of the mean and/or ii) an alternative criterion like Watson's U^2 permutation test. These analyses would not have to be included in the paper, but would lend considerable credibility to these novel and very central findings.

2. Related to the histograms in Fig. 3, I would appreciate if the authors provided more details about the absolute numbers of SO_spindle events used in this analysis. If I read this correctly (and please correct me in case I did not), the SO_spindle events are more or less a subset of the entirety of SO events shown in Fig. 2 – namely those who co-occurred with a spindle within a defined time frame.

Fittingly, the histograms of SO in Fig. 2 and SO_spindles in Fig. 3 are virtually identical (the topographies are similar as well). The authors should report absolute event numbers for the reader to have at least a rough estimate of i) how reliable the SO_spindle result is (i.e., is there a sufficient number of events remaining?) and ii) whether the SO_spindle effect is to be regarded as independent from the previous SO findings or an (almost trivial) extension of the SO effect shown in Fig. 2.

3. Although the figures are certainly high-quality overall, I felt the authors are rather selective in illustrating data in greater depth. For example, Fig. 2a and 2d are restricted to an exemplary electrode (with Fig. S1 providing a little more detail). Presumably for consistency across figures, Fig. 3a also visualises the polar histogram for electrode F2 – here, however, this electrode appears to be an unusual choice since its neighbouring electrodes are marked as non-significant and the topography suggests that the effect might be strongest at other (parietal?) locations.

Moreover, given the broad topography in Fig. 3a, I wonder whether there is additional information to be gained, for example by plotting a topography of preferred phases (potentially using a more robust measure than the circular mean). This would conceivably tell us something about an organisational principle of SO_spindle complexes across the respiratory cycle (in the sense of a 'spatial phase-locking gradient'). Relatedly, it would be very interesting to statistically compare the preferred phases of SO and spindle events (Fig. 2d) with a (non-parametric) dependent-samples test – is there a meaningful phase shift between these two event types on the group level?

4. As a final comment, the 'frontal cluster' shown in Fig 3b comprises (at least) electrode Fz which is marked as non-significant (i.e., not meaningfully related to respiration) in panel a. Could the authors comment as to how, in their opinion, this impedes the full-circle link between respiration, SO_spindle complexes, and memory reactivation?

Minor comments

1. The Abstract should state the $N = 20$

2. It would be helpful to give (rough) frequency ranges for SOs, spindles, and ripples in the Introduction

3. Which electrode is shown in the top panel of Fig. 1c? It would also be helpful to include the electrode (Cz) and colour code information directly into Fig. 1d. Why was Cz chosen here?

4. I strongly suggest to use either violin or raincloud plots instead of the bar graph in Fig. 3a to clearly illustrate bin-wise distributions. Please also state what the error bars show (SD/SEM?).

5. Admittedly a picky comment, but including negative modulation indices to determine critical Z values is not meaningful (H_0 assuming no modulation in the null distribution of spectra).

Reviewer #2 (Remarks to the Author):

This study in healthy humans shows that the occurrence of slow oscillations (SO), spindles and slow oscillation-spindle complexes during sleep increases around the inhalation peak of the breathing rhythm. Moreover, coupling strength between the breathing rhythm and SO-spindle complexes was found to be positively correlated with memory reactivations as determined in the same EEG data set in a previous publication.

The authors report a highly interesting observation suggesting that breathing during NonREM sleep might entrain EEG-oscillations known to be involved in memory consolidation during sleep. The methods appear to be overall sound and the manuscript is written in a comprehensive manner. Nevertheless, some questions should be addressed in a revision of this work:

1) To what extent is the effect of breathing specific to the slow oscillation and spindle rhythms and their co-occurrence? Figure 1a shows an increase in power around the inhalation peak that covers the whole 0.5-24.5 Hz band shown. On the other hand, at the time of the inhalation peak, the cooccurrence of SO-spindle events is found to be significantly increased even if the number of SO-

spindle complexes is normalized, i.e., determined "relative to all SOs at electrode F2" (unfortunately the ms has neither page nor line numbers). This is basically a well thought out approach. Nevertheless, I do not understand why the normalization is restricted to F2 (in the Methods it seems that it is done for all electrodes). Further analyses would be very helpful that more convincingly demonstrate a specific effect of breathing on SO and spindle oscillations (compared to other EEG oscillations). Alternatively, the outcome of such analyses could reveal that the influence of breathing is non-specific, i.e., enhancing any EEG rhythm that prevails depending on the brain state. Also, this question needs to be addressed in the discussion.

2) A related issue is that the authors demonstrate an increase in SP-spindle complexes around the inhalation peak. However, nothing is said about whether breathing also modulates actual phase coupling between SOs and spindles. Such data should be added even if the outcome is negative.

3) Did the authors calculate separate correlations between reactivation and respiration-SO or respiration-spindle coupling? It would be interesting to see whether respiration-SO/spindle co-occurrence is the main predictor for reactivations.

4) The authors take the findings to suggest that breathing is a "potential" pacemaker for EEG SO and spindles rhythms. The impact of the work would be distinctly enhanced by experiments supporting the idea of a causal influence of breathing on SO and spindle regulation (e.g., by manipulating breathing). Although this might involve some experimental effort, at least such possibilities might be discussed.

5) What are the mediating mechanisms of the effect of breathing? Breathing, amongst others, goes along with a rhythmic activation of sympathetic activity. Thus, rather than a peripheral feedback effect (as the authors seem to argue in the Discussion "Many organs interact with brain rhythms...") the influence of breathing may originate from brain stem breathing regulating centers. Please, discuss.

6) The authors report a positive correlation of the strength of the breathing-SO/spindle coupling with "levels of memory reactivation" as determined by decoding accuracy (published in a previous paper). Please, add whether breathing-SO/spindle coupling strength was also correlated with recall measures assessed in this previous paper.

Minor issues:

- Abstract – the term "coupled" SO_spindel complexes may be misleading as it suggests a kind of phase coupling.

- Abstract – "extent of reactivation". This is an important finding, and the authors may wish to specify what exactly is meant by the "extent" already in the abstract.

- As to the analyses of SOs, spindles and SO/spindle complexes, the number of events detected per participant should be indicated. Also, some of the analyses seem to be based on pooled data across participants, others on an n = individual participants. Which approach was used should be clarified in the results, optimally in the figure legends.

- Related: Are SO/spindle events non-uniformly distributed towards the inhalation peak also within each participant?

- I am not sure whether I see this correctly, but does the significant cluster for the correlation (Fig 3b) comprise an electrode that did not show non-uniform coupling towards the respiration peak across participants (Fig. 3a)? Of course, vector length for individual participants might still be large, however, we won't know whether direction was towards the respiration peak.

- Do the authors have data about to what extent participants engaged in nasal or mouth breathing during the sleep period? If so, please add.

- Of the 20 participants, 17 were female. I am just curious: Do the data essentially change without the male participants. Gender might be discussed in the Discussion section.

Reviewer #3 (Remarks to the Author):

This manuscript builds on a growing literature characterizing factors that govern oscillatory dynamics in electroencephalography (EEG) data, with a specific focus on the potential role of respiratory patterns to sculpt the expression of phase amplitude coupling between two distinct oscillations supporting cognitive functions. While prior work in EEG has explored the role of respiratory cycles on

EEG dynamics during waking, this manuscript explores the potential role of respiratory signals to pace with signature brain oscillations during NREM sleep known to support neuroplasticity and memory consolidation. The authors utilize a dataset that includes respiratory monitoring, scalp EEG, and a sleep-dependent associative memory task during a nap study. They find that slow wave-sleep spindle coupling peaks around respiratory inhalation peaks, and that the topographic peak of this coupling, in part, co-localizes with areas exhibiting evidence of memory reactivation. These findings are potentially illuminating, and identify a new candidate factor governing how sleep supports memory consolidation. Overall, the manuscript is well written, and could potentially offer important findings that would advance the field of sleep-dependent memory and identify novel targets for memory enhancement and assessment. Despite the many conceptual, methodological, and analytical strengths of this study, there remain a few outstanding concerns regarding the interpretation of results and contextual framing of conclusions that limit our ability to recommend this manuscript for publication in its current form. Specific comments and concerns are outlined below.

GENERAL

1. One of our central concerns is the core framing of respiration as a pacemaker for the expression of SOs, spindles, and SO-spindle complexes, as well as in the role played by SO-spindle complexes in the process of sleep-dependent memory consolidation. We have several concerns with this framing throughout the manuscript.

a. In the introduction, respiration is being framed implicitly as a potential explanatory pacemaker for the changes in SO-spindle coupling seen in neurodevelopment and aging. However, the authors do not clarify whether these periods are also accompanied by changes in breathing/respiratory rhythms. This should be explicitly addressed to clarify the hypothesis-driven focus on these interactions..

b. The term “pacemaker” carries strong causal connotations, and it is not clear that either the study design or the analyses described in the manuscript meet the standard of evidence needed to establish a causal link. While the authors have made considerable effort to demonstrate that the associations themselves are statistically significant beyond chance levels, it is not adequately established that there exists a direct temporal sequence in which inhalation peaks consistently precede sleep oscillations, or indeed that any such sequence may not be explained at least in part by other physiological variables in play. Figure 2d may hint at the temporal precedence of inhalation peaks over max-amplitude spindles, but the very same figure may equally be visually interpreted as showing temporal precedence of max-amplitude SOs over inhalation peaks. The circular distributions suggest that even among spindles, a sizable minority of vectors show maximum amplitude during negative phases of the inhalation (i.e. precede the inhalation peak).

c. Although the associative findings are meaningful and insightful, and certainly do not preclude the possibility of causal links, in the absence of either analytical (e.g. Granger causality or transfer entropy), statistical (e.g. structural equation or mediation modeling) or experimental approaches, they fall short of the burden of proof for causal inference in this chicken-and-egg problem. In that respect, it is not clear that the respiratory rhythms are “making” pace, so much as keeping pace with the SOs/SO-spindle complexes. This limitation may merit further discussion, as well as changes to language throughout the manuscript.

d. Additionally, the mechanistic and neurobiological justifications for framing respiratory oscillations as pacemakers are not sufficiently detailed. For instance, under the pacemaker framework, it would be important to offer an explanation for how an inhalation may induce neural expression, and especially how any physiological changes tied to an inhalation may propagate with such speed to thalamocortical and cortical expression of neurophysiology.

e. For the above reasons, the findings should be framed and interpreted primarily as being associative. Moreover, it is possible that it is not respiration per se, but another physiological pacemaker that paces both oscillations and respirations that may be driving both signals downstream. Namely, autonomic output such as heart rate variability are also strongly coupled to breathing, and as such- it is likely that the greater pacemaker in question may actually be the autonomic nervous system itself. Use of the term pacemaker suggests that there is a neurobiological mechanism coupling breathing to sleep oscillations, which this experimental set up is not suited to test. It is equally likely that the ANS is the pacemaker driving respiration and neural dynamics as downstream targets. This

should be addressed in some way. In short, the findings have not been appropriately contextualized within a multidimensional framework of interacting, synchronous, and regulated physiological control systems. Cardiac rhythms, and in particular heart rate variability, are known to be associated with both sleep neurophysiology as well as circulatory physiology both during and outside of sleep, and, in the absence of further analyses, they should at the very least be included in any discussion of pacemaking/pacekeeping during sleep. The role of autonomic-central coupling is consequently also an overlooked point of discussion for this study. The potential mechanistic role of the brainstem, which serves as a true pacemaker for the respiratory system as well as the cardiac system, is not addressed at all within this framework, and also merits addressing at least in the discussion.

COMMENTS SPECIFIC TO RESULTS & METHODS

1. Figure 2a is interpreted as showing evidence for enhanced SO/spindle power around the inhalation peak, but the time-frequency t-map clearly depicts a broadband and diffuse, seemingly non-specific cluster enclosed by the contour. This should be more adequately addressed, as the interpretation may not be entirely accurate. For example, it is possible that the plot in Figure 2a is influenced by some form of movement-related or other form of artifact during inspiration and expiration. This should be addressed. The authors should also clarify why, if they think the PAC frequencies are truly neural signal-related, the focus should be specifically on slow oscillations and sleep spindles when there is clear coupling across theta and alpha frequencies as well.

2. The choice of, and reliance on data from channels Cz in some places and F2 in others is not adequately explained or justified. In the methods subsection entitled "Modulation Index", it is neither immediately apparent, nor adequately justified why the channel Cz was selected for this analysis. Channel F2 is initially considered as a representative example, but is later the sole channel used for the analyses shown in Figure 3. While the distribution of preferred phase of coupling between respiratory rhythms and SOs, spindles, and SO-spindle complexes have been shown in supplementary figure 1, the higher order analyses of cognition appear to be focused purely on F2. The manuscript should spend more text explicitly justifying the logic behind choosing the electrode derivations within each analysis or stick with a consistent electrode across analyses, to minimize the appearance of potential cherry picking.

3. In the topographical plots in Figure 3a and 3b, aside from channel F2, there appears to be a lack of agreement in the regional effects shown. The other 2 channels shown in Figure 3b appear to coincide with regions of null effects in Figure 3a, and this is not adequately explained. In fact, it seems that if viewed superimposed, the two topoplots appear to be approximately inverted. This should be addressed in the interpretation and discussion.

4. Given the focus on respiratory oscillations, the authors do not address whether their cohort was screened for sleep-disordered breathing (SDB, e.g., obstructive sleep apnea). Although the PSQI was used to assess sleep quality in this otherwise healthy group, this instrument is not as well-suited nor as sensitive to SDB as other, more targeted questionnaires such as STOP-BANG or the Berlin Questionnaire. Furthermore, other measurements associated with incidence of SDB, such as body-mass index, neck circumference, or waist-to-hip circumference ratio were neither reported nor modeled as analytical covariates. SDB is known to cause sleep fragmentation, due to which oscillatory expression tends to localize to periods without cessation in breathing, which in turn is associated with changes in SO-spindle coupling in individuals with SDB. Given the multifarious links between SDB and altered neural and respiratory rhythms, as well as associated cardiovascular and cerebrovascular impacts, this is an important limitation to address in the discussion. It is possible that breathing abnormalities in the signal may alter SO-spindle or SO-spindle-respiratory coupling in a way that biased the current analyses. This should be addressed in some form.

5. When describing the memory reactivation results associated with Figure 3b, the authors refer to a prior publication to describe the LDA decoding technique. However, in the interest of clarity, at least a concise (1-2 sentence) elaboration on this approach should be included in the methods section of the current manuscript considering it is a central part of the analysis.

MINOR

1. "SO_spindles" and "SO_spindle complexes" should be replaced with SO-spindles/SO-spindle complexes as needed.
2. In the introduction, there is a reference to "in- and decreases in memory..." which should be changed to increases and decreases.
3. In the methods, in the subsection entitled "Experimental Overview", there appears to be a typographical error – the study start time is indicated as being 12am, but the nap initiation occurred at 1pm. If inaccurate, the start time of midnight should be rectified to noon.
4. In the methods subsection entitled "Modulation Index", "z-transformed" should be replaced with z-scored/z-normalized (as used elsewhere in the manuscript), mainly to avoid confusion with the z-transform, which is an altogether different concept in signal processing.
5. In the methods subsection entitled "SO-spindle rate across respiratory cycle", there is a typographical error – "-pi to pi". Further, since this range is in radians, the step size of 18 degrees should also be converted to and reported in radians.

Reviewer 1

Comment1: Respiration methodology

Some aspects of how the respiratory data were preprocessed and analysed are not uncontested in the field. I felt that three instances in particular deserve discussion and/or a control analysis to rule out contamination of the results by certain choices made in the analysis pipeline:

a) The authors employed Hilbert transform to extract the continuous phase of the respiration time series, presumably a built-in function from the BreathMetrics toolbox (although I could be wrong here). Due to inherent characteristics of human respiration, the Hilbert transform is known not to be ideal for this application, and several alternatives have been developed (single and dual interpolation approaches, protophase, and others). While there is no clear consensus regarding best practice, the many data sets from different labs/modalities on which I have seen Hilbert transform fail (and which I am happy to share with the authors, if desired) raise some scepticism about the potential influence of the phase extraction method. It would be great if the authors could report whether and how they made sure that respiration phase was reliably extracted. As a control, I suggest to repeat a single analysis (e.g., the single-sensor analysis of preferred phase shown in Fig. 2d) using a different phase extraction approach, for example by linearly interpolating between peaks and troughs as well as between troughs and peaks (dual interpolation) and compare the resulting polar histograms.

Response: We want to thank the Reviewer for this insightful point. We closely monitored the outcomes of the Hilbert transform by visual inspection, to ensure the soundness of the obtained phase values. Below we show the average of the inhalation-peak-locked data and the corresponding Hilbert transformed data of a sample participant (a) and a 30 seconds data segment comprising both filtered respiratory data and the Hilbert transform (b). Both plots indicate that the Hilbert transformation provided a reliable estimate of the respiratory phase.

(a) Average of inhalation-peak-locked respiratory data and corresponding Hilbert transform of a sample participant.
(b) 30 seconds data segment comprising both filtered respiratory data and the Hilbert transform.

Nevertheless, we followed the Reviewer's advice and repeated the single-sensor analysis of the preferred coupling phase (Fig. 2d) using a dual interpolation procedure. As shown below results were highly similar to our original outcomes. The preferred phases of the respiration-SO modulation clustered just before the inhalation peak (i.e., 0° ; mean angle: $-12.06^\circ \pm 0.25$, mean vector length = 0.29; $V = 5.71$, $p = 0.035$). Likewise, for spindles a significant non-uniform circular distribution was found (V -test against 0° , $V = 6.85$, $p = 0.015$), with spindle onsets clustering right after the inhalation peak (mean angle at P1: $14.38^\circ \pm 0.25$, mean vector length = 0.35). Finally, we directly compared the outcomes with our original results using the Watson-Williams test. No significant differences became apparent (SO: $F = 0.0013$, $p = 0.97$; Spindles: $F = 0.14$, $p = 0.71$).

We added the outcomes of this analysis to the supplemental (Supplementary Fig. 12) and refer to it in the manuscript (page 20, line 419).

...Then a Hilbert transform was applied, and the instantaneous phase angle was extracted (for outcomes based on extracting respiratory phase using a double interpolation¹ approach see Supplementary Fig. 12)...

Supplementary Figure 12. Preferred coupling based on double interpolation. To obtain continuous respiration phase angles, peaks (inhalation) and troughs (exhalation) were detected in the normalised respiration time course. Phase angles were linearly interpolated from trough to peak ($-\pi$ to 0) and peak to trough (0 to π) to obtain respiration cycles centred around peak inhalation (i.e., phase 0). We determined the respiratory phase during detected SO downstates across participants at electrode F2. A significant non-uniform circular distribution became apparent ($V = 5.71$, $p = 0.035$), with SO clustering just before the inhalation peak (i.e., 0° ; mean angle: $-12.06^\circ \pm 0.25$, mean vector length = 0.29). Next, we assessed the preferred respiration phase for spindle onsets at electrode P1. A significant non-uniform circular distribution was found (V -test against 0° ; $V = 6.85$, $p = 0.015$), with spindle onsets clustering right after the inhalation peak (mean angle at P1: $14.38^\circ \pm 0.25$, mean vector length = 0.35). Finally, we directly compared the outcomes with the original Hilbert transformed data (Fig. 2d) using the Watson-Williams test. No significant differences became apparent (SO: $F = 0.0013$, $p = 0.97$; Spindles: $F = 0.14$, $p = 0.71$).

b) For the computation of the modulation index, what were the constraints used for the 'random shifts', i.e. were (sub-)harmonics of the individual breathing frequency excluded or was the width of possible shifts otherwise restricted to a certain range? As a side note, was it really the case that both time series were shifted or was one kept constant and the other was shifted?

Response: We did not apply any constraints for the phase shifts, as we wanted to generate a maximally conservative null distribution. In addition, we indeed initially shifted both the phase and amplitude providing time-series. We computed the Modulation Index again, keeping the EEG time-series constant while only shifting the phase related time-series (i.e., respiratory signal). As visible below, the major outcome that the respiratory phase impacts EEG amplitudes both in low-frequencies (peaking at 0.5 Hz) and in the spindle range (peaking at 15.5 Hz) remained unchanged. Please note that in response to comment 1 (results & methods) of Reviewer 3, electrode F2 (instead of Cz) was now used to provide EEG amplitude measures. We adapted the overall procedure to compute the Modulation Index as described above.

To assess the significance of the modulation index (MI) the phase time series was randomly shuffled and the MI was computed using the shuffled phase time series and the observed amplitude time series. The procedure was repeated 200 times, resulting in a MI-level reference distribution. The mean and standard deviation across the reference distribution was then used to z-score the MI of the empirical data. Z-values were then transformed into p-values, with a significance threshold of z-values greater or smaller than ± 1.96 . The MI index indicates that the phase of respiration significantly influences the EEG amplitude at electrode F2 with local peaks in the SO (0.5 Hz) and spindle range (15.5 Hz; $z > 1.96$; the red line depicts all significantly modulated frequencies; corrected for multiple comparisons across frequencies).

We updated Fig. 2 and the methods section accordingly (page 17, line 438).

...Modulation Index. Phase amplitude coupling was assessed with the Modulation Index (MI)². To estimate instantaneous phase of the respiration signal, we filtered the continuous respiratory data around 0.25 ± 0.05 Hz two-pass Butterworth bandpass filter). We then extracted instantaneous amplitude data across frequencies between 0.5 and 25 Hz at electrode F2 in steps of 1 Hz. To this end, a two-pass FIR filter (order = three times the lower frequency bound) was used to create 20 equally spaced frequency bins, with centre frequencies ranging from 0.5 to 25 Hz, and with fixed-frequency bandwidths of 1 Hz. The envelope of the Hilbert-transformed band-pass filtered data was then used as amplitude estimate.

To compute the MI (for a given frequency pair), we divided the phase signal into 20 bins, and then, for each bin, computed the mean amplitude for that bin. This yields a distribution of amplitude as

a function of phase. The MI is defined as the Kullback-Leibler distance between that distribution and the uniform distribution (over the same number of bins). To assess the statistical significance of the MI values, *we randomly shifted the phase time series*, and computed the MI using the shifted signal³. We repeated this procedure 200 times, resulting in a MI-level reference distribution. The mean and standard deviation across the reference distribution was then used to *z-score* the MI of the empirical data. Z-values were then transformed into p-values, with a significance threshold of z-values greater or smaller than +/- 1.96...

c) The statistics reported for preferred phase are based on the circular means of polar histograms and corresponding tests (V-test against zero). Circular means work well when (and only when) the underlying distribution has a rather clear unimodal characteristic (especially when tested against a known direction with a V-test); for multimodal distributions, the circular mean may yield artificial results (see e.g. Landler et al., Sci Rep 2021). It is rather hard to guess from Fig. 2d and 3a, but it would be very convincing if the authors could report a quick control, e.g. i) using the circular median instead of the mean and/or ii) an alternative criterion like Watson's U^2 permutation test. These analyses would not have to be included in the paper, but would lend considerable credibility to these novel and very central findings.

Response: We thank the Reviewer for this thoughtful comment. While we refrained from applying the Watson's U^2 permutation test (given that it is designed for independent samples), we modified the V-test using the circular median. As visible below, the statistical outcomes were almost identical to our original approach (SOs: V_{median} across sign. electrodes = 8.83; Spindles: V_{median} across sign. electrodes = 9.22; SO_spindles: V_{median} across sign. electrodes = 14.18).

Comment 2: Related to the histograms in Fig. 3, I would appreciate if the authors provided more details about the absolute numbers of SO_spindle events used in this analysis. If I read this correctly (and please correct me in case I did not), the SO_spindle events are more or less a subset of the entirety of SO events shown in Fig. 2 – namely those who co-occurred with a spindle within a defined time frame. Fittingly, the histograms of SO in Fig. 2 and SO_spindles in Fig. 3 are virtually identical (the topographies are similar as well). The authors should report absolute event numbers for the reader to have at least a rough estimate of i) how reliable the SO_spindle result is (i.e., is there a sufficient number of events remaining?) and ii) whether the

SO_spindle effect is to be regarded as independent from the previous SO findings or an (almost trivial) extension of the SO effect shown in Fig. 2.

Response: We apologize for not having included the number of events (i.e., SOs, spindles, SO-spindles) in the manuscript. While they are reported in the previous publication (Schreiner et al., 2021) we now added a dedicated table to the Supplementary (see below).

The Reviewer is correct that SO-spindle events represent a subset of all detected SO events (which consist of solitary SOs and coupled SO-spindle complexes). It is also true that the temporal modulation of all SOs and SO-spindle complexes by respiration are very similar, but not identical. But it has to be noted that when we assessed whether the likelihood of SOs to group spindles might be modulated by the respiration phase, we binned the number of SO_spindles *relative to all SOs*. In other words, the modulation seen in Fig. 3a (modulation of SO-spindles by respiration) goes beyond the impact of respiration on all SOs, as the distribution would be entirely flat otherwise. This result indicates that respiration not only has an influence on SO-spindle coupling but that SO-spindles might be preferentially affected by respiration.

Sleep stage [%]	MEAN \pm SEM
N1	12.5 \pm 1.5
N2	43.8 \pm 1.6
SWS	21.1 \pm 2.6
REM	19.4 \pm 2.6
WASO	1.9 \pm 0.6
Total Sleep Time [min]	101.6 \pm 2.8
# spindles	184.8 \pm 12.9
# SOs	459.5 \pm 34.5
#SO_spindles	50.1 \pm 3.5

Supplementary Table 1. Sleep characteristics (averaged across conditions ⁴). Data are means \pm s.e.m. N1, N2: NREM sleep stages N1 & N2, SWS: slow-wave sleep, REM: rapid eye movement sleep, WASO: wake after sleep onset.

Comment 3: Although the figures are certainly high-quality overall, I felt the authors are rather selective in illustrating data in greater depth. For example, Fig. 2a and 2d are restricted to an exemplary electrode (with Fig. S1 providing a little more detail). Presumably for consistency across figures, Fig. 3a also visualises the polar histogram for electrode F2 – here, however, this electrode appears to be an unusual choice since its neighbouring electrodes are marked as non-significant and the topography suggests that the effect might be strongest at other (parietal?) locations.

Moreover, given the broad topography in Fig. 3a, I wonder whether there is additional information to be gained, for example by plotting a topography of preferred phases (potentially using a more robust measure than the circular mean). This would conceivably tell

us something about an organisational principle of SO_spindle complexes across the respiratory cycle (in the sense of a 'spatial phase-locking gradient'). Relatedly, it would be very interesting to statistically compare the preferred phases of SO and spindle events (Fig. 2d) with a (non-parametric) dependent-samples test – is there a meaningful phase shift between these two event types on the group level?

Response: We thank the Reviewer for these helpful comments. The Reviewer is right in the assumption that electrode F2 was highlighted in Fig. 3a for consistency. The strongest effect (in terms of V-values) was observed at electrode C4 ($V = 13.34$, $p < 0.0001$). Importantly, the direction of the effect was identical to the one shown in the main figure (at electrode F2), indicating that SO_spindles preferentially emerged just before the inhalation peak.

We decided now to show not only the average of all significant electrodes in the Supplemental (Supplementary Fig. 10) but also the electrodes exhibiting the strongest effects (Supplementary Fig. 11; also see below). We refer to the Figures in the Methods section, also outlining the rationale of electrode choice (page 15, line 376):

... We used electrode F2 as representative electrode for all SO and SO_spindle related analyses and electrode P1 in case of spindles. In Supplementary Fig. 10 we highlight the average of all significant electrodes for phase-related analyses and in Supplementary Fig. 11 we show the electrodes exhibiting the strongest phase-modulation related effects...

Supplementary Figure 11. Electrodes exhibiting the strongest preferred coupling. The strongest preferred coupling between SOs and respiration (in terms of V-values) became apparent at electrode F2 (red: mean angle: $-9.4^\circ \pm 0.22$; V-test against 0° , $V = 10.87$, $p < 0.001$). In case of spindles the strongest preferred coupling was located at electrode CP1 (blue: mean angle: $16.95^\circ \pm 0.23$; V-test against 0° , $V = 9.99$, $p < 0.001$). For SO_spindles electrode the strongest modulation by respiration was detectable at electrode C4 (green: mean angle: $-4.5^\circ \pm 0.16$; V-test against 0° , $V = 13.34$, $p < 0.001$).

In addition, as suggested by the Reviewer, we computed the topography of preferred phases for SO_spindle modulation by respiration (both based on the circular mean and the circular median). The preferred phase topographies illustrate that SO_spindles tended to occur briefly before the inhalation peak (i.e., 0°) across the cluster where significant phase modulation was detectable (as illustrated in Fig. 3a). We added the figure to the Supplemental and refer to it in the manuscript (page 8; line 171).

...for a topography of preferred phases for SO_spindle modulation by respiration see Supplementary Fig. 8...

Supplementary Figure 8. Topographies of preferred phases for SO-spindle modulation by respiration (left: based on circular mean; right: based on circular median). The topographies illustrate that SO_spindles tended to occur briefly before the inhalation peak (i.e., 0°) across the cluster where significant phase modulation was detectable (as shown in Fig. 3a).

Finally, we followed the Reviewer's suggestion and statistically compared the preferred phases of SO and spindle events (Fig. 2d) to assess whether there is a meaningful phase shift between these two event types at the group level. It has to be noted that our main approach of locking spindle related analyses to the onset of spindles does not favour this question, given that spindles tended to emerge briefly after the inhalation peak, hence in close proximity to SO-down-states (which clustered briefly before the inhalation peak). Hence, we assessed in a first step again the preferred respiration phase for spindles, but this time with regards to spindle peaks. As shown below, similar to spindle onsets, spindle peaks clustered after the inhalation peak (V-test against 0° , mean $V = 5.74 \pm 0.25$, $p < 0.05$, corrected; mean angle of significant electrodes: $50.45^\circ \pm 0.21$, mean vector length = 0.57)

Next, we assessed using the circular Watson-Williams test whether there would be a phase shift between SOs and spindles in relation to their modulation by respiration. We found a significant phase shift across a broad cluster (all $p < 0.05$; corrected) indicating that SOs and spindles clustered at different phases of the respiratory cycle, with SOs emerging before the inhalation peak and spindles following.

We added the Figure to the Supplemental and refer to it in the main text (page 6, line 125).

...Together, these results reveal a strong modulation of SOs and spindles by respiration (see Supplementary Fig. 2 indicating a phase shift between SOs and spindles in relation to their modulation by respiration)...

-- Preferred phase [Spindle peak - respiration coupling] --

--- Comparison of preferred phases in relation to SO & spindle events ---

Supplementary Figure 2. (Top) To assess whether there is a phase shift between SOs and spindles in relation to their modulation by respiration, we assessed in a first step the preferred respiration phase for spindle coupling with regards to spindle peaks (the original analysis reported in Fig. 2d with regards to spindle onsets impedes such an analytical procedure due to the proximity of spindle onsets to inhalation peaks and hence SO down-states). Similar to spindle onsets, spindle peaks clustered after the inhalation peak (V-test against 0° , mean $V = 5.74 \pm 0.25$, $p < 0.05$, corrected; mean angle of significant electrodes: $50.45^\circ \pm 0.21$, mean vector length = 0.57). (Bottom) Next, we assessed – using the circular Watson-Williams test - whether there would be a phase shift between SOs (i.e., downstates) and spindles (i.e., peaks) in relation to their modulation by respiration. We found a significant phase shift across a broad cluster (all $p < 0.05$; corrected) indicating that SOs and spindles clustered at different phases of the respiratory cycle, with SOs emerging before the inhalation peak and spindles following.

Comment 4: As a final comment, the ‘frontal cluster’ shown in Fig 3b comprises (at least) electrode Fz which is marked as non-significant (i.e., not meaningfully related to respiration) in panel a. Could the authors comment as to how, in their opinion, this impedes the full-circle link between respiration, SO_spindle complexes, and memory reactivation?

Response: In the course of the revision we found a slight mistake in the illustration of the SO_spindle – respiration modulation (Fig. 3a). We realized that the figure was erroneously based on the non-collapsed 40 datasets of our 20 participants. We corrected this mistake, by collapsing across both sessions per participant before statistical assessment and plotting, as it was done in all of the other analyses (please note that only the illustration of the topography was affected, without any other major differences; please see below). We updated Fig. 3a and apologize for the mistake.

That said, the Reviewer is correct that one electrode (FC2) of the significant cluster in Fig. 3b (coupling-decoding correlation) did not show a significant non-uniform distribution with regards to SO_spindle – respiration coupling (Fig. 3a). However, even though the distribution at FC2 did not reach significance ($V = 5.11$, $p = 0.052$) and therefore did not enter the cluster, SO-spindles tended to group around the inhalation peak (mean direction: $1.4 \pm 0.21^\circ$; vector length: 0.25), indicating that results at electrode FC2 were not qualitatively different.

-- SO_spindle - respiration coupling [FC2] --

Minor comments

1. The Abstract should state the $N = 20$

Response: We added the information to the abstract.

...We recorded scalp EEG and respiration throughout an experiment in which participants ($N = 20$) acquired associative memories before taking a nap...

2. It would be helpful to give (rough) frequency ranges for SOs, spindles, and ripples in the Introduction

Response: We added the requested information to the Introduction (page 3, line 25).

...This hippocampal-cortical communication is thought to be facilitated by the multiplexed co-occurrence of cardinal non-rapid eye movement (NREM) sleep oscillations, namely cortical slow oscillations (SOs; ~ 1 Hz), thalamic sleep spindles ($\sim 12-16$ Hz), and hippocampal ripples ($\sim 80-120$ Hz in humans)^{5,6}...

3. Which electrode is shown in the top panel of Fig. 1c? It would also be helpful to include the electrode (Cz) and colour code information directly into Fig. 1d. Why was Cz chosen here?

Response: We added all relevant information to Fig. 1. As in 1d EEG for the raw trace in 1c was extracted from channel Cz. Cz was used for illustrational purposes as it represents a common *a priori* region of interest in sleep EEG work, as fast spindles and SOs tend to coincide there (Ngo & Staresina, 2022; Schreiner et al. 2021; Hahn et al, 2020; Muehlroth et al., 2020 etc.).

Fig. 1 Experimental procedure. (a) During encoding, participants were presented with 120 verb-object or verb-scene pairs (counterbalanced across sessions). Memory performance was tested before and after a 120 min nap period. Each session ended with a localizer task in which participants processed a new set of object and scene images. (b) Hypnogram of a sample participant, showing time spent in different sleep stages across one nap. The grey shading indicates NREM sleep stages N2 and SWS. (c) Example of a NREM sleep segment at Cz (30 s; top row: EEG recording; bottom row: respiration). (d) Example of 1/f corrected power spectrum during NREM sleep at Cz for EEG (red) and respiratory recordings (blue), as obtained by Irregular Resampling Auto-Spectral Analysis (IRASA)⁷.

4. I strongly suggest to use either violin or raincloud plots instead of the bar graph in Fig. 3a to clearly illustrate bin-wise distributions. Please also state what the error bars show (SD/SEM?).

Response: We believe that while both violin and raincloud plots are great ways to illustrate data distributions under certain circumstances, they would impede the comprehensibility of our results. However, to comply with the Reviewer's suggestion to illustrate bin-wise distributions, we decided to additionally use boxplots displaying the median, the lower and upper quantiles and the minimum and maximum values. We added the Figure to the Supplemental and refer to it in the figure legend of Figure 3.

...In all participants, *SO_spindle rate (relative to SOs ± SEM) at F2 was non-uniformly distributed across the respiration cycle (all $p < 0.001$; corrected, see right panel; for an alternative illustration showing the bin-wise distributions see Supplementary Fig. 8)*...

-- SO_spindle rate [relative to SOs] --
across respiration

Supplementary Figure 8. Illustrating bin-wise distributions of the modulating effect of respiration on SO_spindle rate (relative to SOs \pm SEM) at F2 (as displayed in Fig. 3a). The black horizontal line represents the median, turquoise boxes the lower and upper quantiles and vertical black lines the minimum and maximum values.

5. Admittedly a picky comment, but including negative modulation indices to determine critical Z values is not meaningful (H0 assuming no modulation in the null distribution of spectra).

Response: We agree with the Reviewer, but as illustrated below the shuffled distributions did not contain any negative values.

Reviewer 2:

Comment 1: To what extent is the effect of breathing specific to the slow oscillation and spindle rhythms and their co-occurrence? Figure 1a shows an increase in power around the inhalation peak that covers the whole 0.5-24.5 Hz band shown. On the other hand, at the time of the inhalation peak, the cooccurrence of SO-spindle events is found to be significantly increased even if the number of SO-spindle complexes is normalized, i.e., determined “relative to all SOs at electrode F2” (unfortunately the ms has neither page nor line numbers). This is basically a well thought out approach. Nevertheless, I do not understand why the normalization is restricted to F2 (in the Methods it seems that it is done for all electrodes). Further analyses would be very helpful that more convincingly demonstrate a specific effect of breathing on SO and spindle oscillations (compared to other EEG oscillations). Alternatively, the outcome of such analyses could reveal that the influence of breathing is non-specific, i.e., enhancing any EEG rhythm that prevails depending on the brain state. Also, this question needs to be addressed in the discussion.

Response: We thank the Reviewer for these excellent comments. The Reviewer is correct that the time-frequency representation in Fig. 2a exhibits inhalation-locked power increases across a broad range of frequencies. However, as elaborated below, the data suggest that effects are indeed driven by SOs and (fast) sleep spindles, with effects in other frequency bands resulting from frequency smearing.

First, we realized that our way of illustrating the obtained effects, i.e., plotting the summed t-values, blurs the fact that the significant cluster comprises two distinct frequency ranges: one spanning slow frequencies (< 10 Hz) and one spanning faster frequencies (14-25 Hz). This becomes more apparent when plotting the mean of z-values across all significant electrodes instead of the summed t-values (please see below - we updated Fig. 1a accordingly). Importantly, oscillatory power associated with SOs and sleep spindles is usually not solely confined to the classical frequency range of <1 Hz (SOs) and 12-16 Hz (spindles). This might reflect slight variations in centre frequencies across individual events or participants, or simply the spectral smoothing inherent in TFR analyses (for corresponding effects see e.g., Cox et al., 2014; Weber et al., 2021; Schreiner et al., 2021). Importantly, we used a complementary approach that uses discrete, algorithmically detected SOs (0.3 – 1.25 Hz) and spindles (12-18 Hz) for determining the preferred phase of the respiration-SO and respiration-spindle coupling, respectively (Figure 2D). The results of these analyses likewise indicate that SOs and sleep spindle activity culminated just before (in case of SOs) and right after the inhalation peak (in case of spindles).

Time-frequency representation of NREM sleep EEG data locked to inhalation peaks, contrasted against randomly selected, matched data segments (mean of z-values across all significant electrodes). The contour lines indicate significant clusters ($p < 0.001$, corrected), illustrating enhanced power in the SO_spindle range around the inhalation peak (time = 0). The white waveform depicts the inhalation-peak-locked respiration signal (mean \pm SEM across participants).

That said, to directly test the hypothesis that the influence of breathing might be non-specific, i.e., enhancing any EEG rhythm prevalent in a given brain state, we assessed whether breathing would also impact slow spindles (9-12 Hz; Mölle et al., 2012) during NREM sleep. Hence, we detected slow spindles in the data (for an illustration of the averaged slow spindle ERP at electrode Fz and the topography of the slow spindle density see below) and then assessed whether slow spindles would cluster at a preferred phase with regards to respiration (just as it was done for fast spindles in the original analysis, see Fig. 1d). We found no significant non-uniform circular distribution across electrodes, indicating that slow spindles are putatively not modulated by respiration (please see below). This is an important indication that not all NREM sleep related rhythms are equally influenced by breathing. We added the information to the results section (page 7, line 128).

...An important question is whether the impact of respiration during NREM sleep is specific to SOs and (fast) spindles, or whether respiration impacts any EEG rhythm that prevails depending on the brain state. We addressed this topic by assessing whether slow spindles (9-12 Hz³¹) would be similarly modulated by respiration. Determining the respiratory phases during the onset of slow spindles across participants revealed no significant non-uniform circular distribution (all $p > 0.05$; see Supplementary Fig. 3), indicating that that the emergence of slow spindles was not robustly impacted by respiration...

Supplementary Figure 3. Preferred coupling phase for slow spindles. (left) Grand average EEG trace of slow spindles at electrode Fz. (middle) Topographic distribution of slow spindle density [events / minute]. (left) Determining the respiratory phases during the onset of slow spindles across participants reveals no significant non-uniform circular distribution (all $p > 0.05$; displayed in the insert is the electrode with the highest V-value: F3, V-value = 4.7; $p = 0.065$).

We also added a dedicated section to the discussion elaborating on the question whether the influence of breathing might be non-specific (page 11, line 262).

...Does respiration impact any EEG rhythm that prevails during a given brain state? Our result that slow spindles, which are dominant over frontal areas and emerge preferentially at the transition into the SO down-state³¹, were not robustly impacted by respiration (Supplementary Fig. 3) points to a specific effect on SOs and fast spindles...

Moreover, as suggested by the Reviewer, we assessed whether the likelihood of SOs to group spindles is modulated by the respiration phase across electrodes. We binned the number of SO_spindles relative to all SOs at each electrode across the respiration cycle in 20 evenly spaced bins (i.e., from $-\pi$ to π). Next, we averaged the distributions per electrode across participants and assessed (non)uniformity of the resulting distribution using the Kolmogorov-Smirnov test. Non-uniformity peaked at frontal, central and parietal electrodes indicating that the likelihood of SOs to group with spindles was robustly modulated by respiration. We added the information to the Supplemental and refer to it in the manuscript (page 8, line 179).

...We found non-uniform distributions in all participants (all $p < 0.001$; corrected for multiple comparisons across participants using FDR correction; see Fig. 3a), indicating a preferential modulation of coupled SO_spindle events by respiration phase (see Supplementary Fig. 5 for results across electrodes...

Supplementary Figure 5. Modulation of the likelihood of SOs to group spindles by the respiration phase across electrodes. The number of SO_spindles relative to all SOs at each electrode was binned across the respiration cycle in 20 evenly spaced bins (i.e., from $-\pi$ to π in steps of 0.31 radians. Next, the distributions were averaged per electrode across participants and (non)uniformity of the resulting distribution was assessed using the Kolmogorov-Smirnov test. Non-uniformity peaked at frontal, central and parietal electrodes indicating that the likelihood of SOs to group with spindles was robustly modulated by respiration.

Comment 2: A related issue is that the authors demonstrate an increase in SP-spindle complexes around the inhalation peak. However, nothing is said about whether breathing also modulates actual phase coupling between SOs and spindles. Such data should be added even if the outcome is negative.

Response: We thank the Reviewer for this great suggestion. To assess whether the consistency of SO-spindle coupling is modulated by the respiration phase, we determined in each participant the respiratory peak frequency and filtered the respiratory data (locked to inhalation peaks) around the peak frequency (± 0.05 Hz, two-pass Butterworth bandpass filter, order = three cycles of the low frequency cut-off). Next, we binned the SO-spindle data across the respiration cycle in 20 evenly spaced bins and computed the vector length, hence the strength of SO-spindle phase-amplitude coupling at electrode F2 per bin. As shown below, SO-spindle coupling was moderately impacted by respiration, with the highest vector length around the inhalation peak (i.e., 0°). Again, (non)uniformity of the resulting distribution was assessed per participant using the Kolmogorov-Smirnov test. We found non-uniform distributions in all participants (all $p < 0.01$; corrected for multiple comparisons across participants using FDR correction). We added the Figure to the Supplemental and refer to it in the manuscript (page 8, line 180).

...see Supplementary Fig. 6 for results with regards to the impact of respiration on the consistency of SO-spindle coupling...

Supplementary Figure 6. To assess whether the consistency of SO-spindle coupling is modulated by the respiration phase, we determined in each participant the respiratory peak frequency and filtered the respiratory data (locked to inhalation peaks) around the peak frequency (± 0.05 Hz, two-pass Butterworth bandpass filter, order = three cycles of the low frequency cut-off). Then a Hilbert transform was applied, and the instantaneous phase angle was extracted. Next, we binned the SO-spindle data across the respiration cycle in 20 evenly spaced bins and computed the vector length, hence the amount of SO-spindle phase-amplitude coupling at electrode F2. (Non)uniformity of the resulting distribution was assessed per participant using the Kolmogorov-Smirnov test. We found non-uniform distributions in all participants (all $p < 0.01$; corrected for multiple comparisons across participants using FDR correction), indicating that SO-spindle coupling was moderately impacted by the respiratory phase with the highest vector length around the inhalation peak (i.e., 0° , bars reflect the average vector length \pm SEM per bin).

Comment 3: Did the authors calculate separate correlations between reactivation and respiration-SO or respiration-spindle coupling? It would be interesting to see whether respiration-SO/spindle co-occurrence is the main predictor for reactivations.

Response: Conducting robust regressions between levels of memory reactivation and respiration-SO coupling or respiration-spindle coupling did not yield any significant effect (all $p > 0.05$ after correcting for multiple comparisons; see below).

Coupling strength [SO-respiration, spindle-respiration] and memory reactivation. Results of a robust regression exhibiting no significant positive relationship between the respiration – SO coupling strength (i.e., vector length) and levels of memory reactivation (all $p > 0.2$) and between respiration – spindle coupling strength and levels of memory reactivation (all $p > 0.1$).

Comment 4: The authors take the findings to suggest that breathing is a “potential” pacemaker for EEG SO and spindles rhythms. The impact of the work would be distinctly enhanced by experiments supporting the idea of a causal influence of breathing on SO and spindle regulation (e.g., by manipulating breathing). Although this might involve some experimental effort, at least such possibilities might be discussed.

Response: The Reviewer is correct that our results are correlational in nature and do not provide evidence for any causal influence of breathing on the timing of sleep oscillations. That said, we now assessed whether respiration drives activity in the SO and sleep spindle range or vice versa using the phase-slope index (PSI, Jiang et al., 2015). The PSI is a metric used to infer the causal direction of oscillatory interactions from electrophysiological time series (Nolte et al., 2010). The cross-frequency PSI was calculated between the respiration signal (filtered around the peak frequency ± 0.05 Hz) and electrode F2 in case of SOs (filtered around 0.3 to 2 Hz) and electrode P1 in case of spindles (filtered around 12 and 18 Hz). In this context, positive values indicate respiration driving SO /sleep spindle activity, while negative values indicate sleep spindles / SOs driving respiration. The obtained data distributions were tested against zero, using paired samples t-tests.

We found that respiration predicted both SO and spindle activity, as evidenced by a positive PSI (SOs: $t_{1,19} = 2.53$; $p = 0.02$; spindles: $t_{1,19} = 2.7$; $p = 0.014$). We added the information to the Supplemental and refer to it in the manuscript (page 6, line 112 for SOs; page 6, line 122 for spindles).

...For results concerning the directional influence of respiration on SOs (indicating that respiration predicted SO activity), see Supplementary Fig. 1...

...For results concerning the directional influence of respiration on spindles (indicating that respiration predicted spindle activity), see Supplementary Fig. 1...

Supplementary Figure 1. Directional influence of respiration on SO and spindle activity. We assessed whether respiration drives activity in the SO and sleep spindle range or vice versa using the phase-slope index (PSI)⁸. The cross-frequency PSI was calculated between the respiration signal (filtered around the peak frequency ± 0.05 Hz) and electrode F2 in case of SOs (filtered around 0.3 to 2 Hz) and electrode P1 in case of spindles (filtered around

12 and 18 Hz). In this context, positive values indicate respiration driving SO /sleep spindle activity, while negative values indicate sleep spindles / SOs driving respiration. The obtained data distributions were tested against zero, using paired samples t-tests. We found that respiration predicted both SO and spindle activity, as evidenced by a positive PSI (SOs: $t_{1,19} = 2.53$, $p = 0.02$; spindles: $t_{1,19} = 2.7$; $p = 0.014$).

Moreover, as suggested by the Reviewer we added a dedicated section to the discussion delineating potential experiments that might support the idea of a causal influence of breathing on sleep oscillations in future studies (page 12, line 287).

...On that note, it deserves mention that our results are correlational in nature. Hence, future studies, directly manipulating breathing behaviour, will be needed to provide causal evidence for the role of breathing in modulating sleep related oscillations. For example, it has been shown that presenting odours during NREM sleep is capable of modifying respiration during sleep, by decreasing inhalation and increasing exhalation volume for up to 6 breaths⁶⁴. One hypothesis would be that such interference (i.e., shallower inhalation due to odour presentation) might mitigate the modulation of sleep related oscillations during the initial presence of odours. Another line of research potentially providing causal evidence could stem from patients suffering from obstructive sleep apnoea (OSA). OSA is characterized by repetitive upper airway collapse resulting in intermittent hypoxia⁶⁵. Strikingly, in OSA patients' sleep seems to have lost its beneficial effect on memory⁶⁶. While it has been shown that abnormal sleep spindle properties are associated with OSA⁶⁷, it remains unclear whether OSA also impacts the emergence of SOs and SO_spindle complexes. This would corroborate the notion that pathological, arrhythmic breathing affects the coordination of sleep oscillations. Importantly, showing that restoring normal breathing behaviour (e.g., via Continuous Positive Airway Pressure (CPAP) treatment) also restores the influence of respiration on sleep oscillations would lend some causal evidence for the role of breathing on the coordination of sleep oscillations...

Comment 5: What are the mediating mechanisms of the effect of breathing? Breathing, amongst others, goes along with a rhythmic activation of sympathetic activity. Thus, rather than a peripheral feedback effect (as the authors seem to argue in the Discussion "Many organs interact with brain rhythms...") the influence of breathing may originate from brain stem breathing regulating centers. Please, discuss.

Response: We added a dedicated section to the discussion outlining how breathing might influence brain activity (page 11; line 266).

...How might breathing exert its impact on brain activity? When we inhale, the incoming airflow stimulates mechanoreceptors of the olfactory sensory neurons⁵⁶. These in turn produce breathing-locked oscillations, which are conveyed to the olfactory bulb and further to the olfactory cortex⁵⁷. The olfactory cortex, however, is not the last terminal of breathing-entrained rhythms. The impact of breathing on neuronal activity has been identified in rodent models and humans in various brain areas^{20-22,40,41,43,44,58}. This anatomical circuitry is in line with recent findings indicating a privileged role of nose breathing as compared to mouth breathing in synchronizing neuronal oscillations and affecting memory processes^{17,39}. However, the brainstem houses respiratory rhythm generators, which might likewise account for the breathing- related entrainment of neuronal oscillations^{43,59} (irrespective of nose or mouth breathing). Specifically, the phases of the respiratory cycle, comprising inhalation and exhalation, are governed by brainstem circuits⁵⁹. The brainstem, as major control hub of the autonomous nervous system, likewise impacts the activity of other vital functions such as the heart rate

or gastric functions^{60,61}. Interestingly, cardiac activity and respiration are intimately linked⁶², while cardiac rhythms and in particular heart rate variability have been shown not only to covary with different sleep stages but also with SO and sleep spindle activity during NREM sleep⁶³. Hence, the extent to which sensory (i.e., olfactory bulb route) or non-sensory (i.e., brainstem route) breathing-locked inputs drive neuronal activity and whether they innervate the same target regions in the brain remains unclear. Controlling the breathing route during sleep would permit drawing more causal inferences whether nose breathing is indeed key for clocking neuronal activity during sleep....

Comment 6: The authors report a positive correlation of the strength of the breathing-SO/spindle coupling with “levels of memory reactivation” as determined by decoding accuracy (published in a pervious paper). Please, add whether breathing-SO/spindle coupling strength was also correlated with recall measures assessed in this pervious paper.

Response: To address this important question, we conducted, at each electrode, a robust regression across participants between (i) the respiration-SO_spindle coupling strength (i.e., vector length) and (ii) the behavioural levels of associative memory consolidation (i.e., proportion of post-sleep recalled images (out of hits) in relation to pre-sleep memory performance as reported in Schreiner at al., 2021). While the outcomes of the procedure did not survive correction for multiple comparisons (FDR), it became apparent that the association between respiration-SO_spindle coupling and memory consolidation was strongest at electrode F2 ($F = 3.56$, $p = 0.075$), mirroring the relationship between coupling strength and reactivation levels (Fig. 3b). We added this result to the Supplemental and refer to it in text (page 9, line 190).

...As shown in Fig. 3b, we observed a significant positive relationship between the two variables at frontal electrodes ($p < 0.05$, corrected for multiple comparisons across electrodes using FDR⁹; for the relationship between respiration-SO_spindle coupling strength and behavioural levels of memory consolidation see Supplementary Fig. 7)...

Supplementary Figure 7. Coupling strength and memory consolidation. We conducted, at each electrode, a robust regression across participants between (i) the respiration-SO_spindle coupling strength (i.e., vector length) and (ii) the behavioural levels of associative memory consolidation (i.e., proportion of post-sleep recalled images (out of hits) in relation to pre-sleep memory performance as reported in¹³). While the outcomes of the procedure did not survive correction for multiple comparisons (FDR), it became apparent that the association between respiration-SO_spindle coupling and memory consolidation was strongest at electrode F2 ($F = 3.56$, $p = 0.075$), mirroring the relationship between coupling strength and reactivation levels (Fig. 3b).

Minor issues:

- Abstract – the term “coupled” SO_spindle complexes may be misleading as it suggests a kind of phase coupling.

Response: We removed the word “coupled”.

- Abstract – “extent of reactivation”. This is an important finding, and the authors may wish to specify what exactly is meant by the “extent” already in the abstract.

Response: Thanks for this hint. We explain now what is meant with “extent” (page 2, line 12).

... Moreover, the strength of respiration-SO_spindle coupling is linked to the extent of memory reactivation (i.e., classifier evidence in favour of the previously learned stimulus category) during SO_spindles...

- As to the analyses of SOs, spindles and SO/spindle complexes, the number of events detected per participant should be indicated. Also, some of the analyses seem to be based on pooled data across participants, others on an n = individual participants. Which approach was used should be clarified in the results, optimally in the figure legends.

Response: We added now a table to the Supplemental reporting (i) the amount of sleep spent in different sleep stages and the number of detected events (see Supplementary table 1).

Sleep stage [%]	MEAN ± SEM
N1	12.5 ± 1.5
N2	43.8 ± 1.6
SWS	21.1 ± 2.6
REM	19.4 ± 2.6
WASO	1.9 ± 0.6
Total Sleep Time [min]	101.6 ± 2.8
# spindles	184.8 ± 12.9
# SOs	459.5 ± 34.5
#SO_spindles	50.1 ± 3.5

Supplementary Table 1. Sleep characteristics (averaged across conditions ⁴). Data are means ± s.e.m. N1, N2: NREM sleep stages N1 & N2, SWS: slow-wave sleep, REM: rapid eye movement sleep, WASO: wake after sleep onset.

Concerning the statistical procedures: All procedures comprised group level statistics with the only exception constituting the assessment whether the likelihood of SOs to group spindles is modulated by the respiration phase (Fig. 3a). Here we evaluated the non-uniformity of the distributions using the Kolmogorov-Smirnov test within each participant.

This information is trackable in

- the results section (page 8, line 175):

...(Non)uniformity of the resulting distribution was assessed per participant using the Kolmogorov-Smirnov test¹⁰. We found non-uniform distributions in all participants (all $p < 0.001$; corrected for multiple comparisons across participants using FDR correction; see Fig. 3a), indicating a preferential modulation of coupled SO_spindle events by respiration phase...

- the figure legend (page 9, line 198):

... In all participants, SO_spindle rate (relative to SOs \pm SEM) at F2 was non-uniformly distributed across the respiration cycle (all $p < 0.001$; corrected, see right panel)...

- and the Methods (page 19, line 513):

*...To assess (non)uniformity of SO_spindle rate across the respiration cycle (Fig. 3 a) the Kolmogorov-Smirnov test was applied **within each participant**...*

- Related: Are SO/spindle events non-uniformly distributed towards the inhalation peak also within each participant?

Response: For SOs we found on an individual level significant non-uniform distributions in 15/20 participants, while 18/20 participants exhibited non-uniform distributions with regards to spindles. Lastly, 18/20 participants exhibited non-uniform distributions with regards to SO_spindles. We added this information to the manuscript.

SOs (page 6, line 111):

...On an individual level we found significant non-uniform distributions in 15/20 participants...

spindles (page 6, line 121):

...On an individual level we found significant non-uniform distributions in 18/20 participants...

SO_spindles (page 8, line 170):

...On an individual level we found significant non-uniform distributions in 18/20 participants (for a topography of preferred phases for SO_spindle modulation by respiration see Supplementary Fig. 4)...

- I am not sure whether I see this correctly, but does the significant cluster for the correlation (Fig 3b) comprise an electrode that did not show non-uniform coupling towards the respiration

peak across participants (Fig. 3a)? Of course, vector length for individual participants might still be large, however, we won't know whether direction was towards the respiration peak.

Response: The Reviewer is correct that one electrode (FC2) of the significant cluster in Fig. 3b (coupling-decoding correlation) did not show a significant non-uniform distribution with regards to SO_spindle – respiration coupling (Fig. 3a). However, even though the distribution at FC2 did not reach significance ($V = 5.11$, $p = 0.052$), SO-spindles tended to cluster towards the inhalation peak (mean direction: $1.4 \pm 0.21^\circ$; vector length: 0.25), indicating that results at electrode FC2 were not qualitatively different.

-- SO_spindle - respiration coupling [FC2] --

That said, in the course of the revision we found a mistake in the illustration of the SO_spindle – respiration modulation (Fig. 3a). We realized that the figure was erroneously based on the non-collapsed 40 datasets of our 20 participants. We corrected this mistake, by collapsing across both sessions per participant before statistical assessment and plotting, as it was done in any of the other analyses (please note that only the illustration of the topography was affected and no major differences became apparent; please see below). We updated Fig. 3a and apologize for the mistake.

- Do the authors have data about to what extent participants engaged in nasal or mouth breathing during the sleep period? If so, please add.

Response: We recorded breathing using a using an Embla thermistor airflow sensor. This device integrates both nose and mouth breathing. Consequently, we cannot draw any conclusions to what extent our participants engaged in nasal or oral breathing during sleep, respectively.

However, oral respiration reflects only ~4% of overall respiration during sleep in healthy participants (Fitzpatrick et al. 2003), suggesting that our results were putatively driven by nasal breathing.

- Of the 20 participants, 17 were female. I am just curious: Do the data essentially change without the male participants. Gender might be discussed in the Discussion section.

Response: We conducted all major analyses again excluding the datasets of the three male participants (please see below). We found almost identical patterns of result indicating that (i) respiration modulated the emergence of sleep oscillations (SOs, spindles and SO_spindles), while (ii) the strength of respiration-SO_spindle coupling was linked to the extent of memory reactivation. We added this information to the Supplementary and refer to it in text (page 14, line 339).

...In brief, twenty healthy participants (mean age: 20.75 ± 0.35 ; 17 female) took part in the experiment (for results comprising only female participants see Supplementary Fig. 9)...

Supplementary Figure 9. Results comprising data from female participants only. (a) Time–frequency representation of NREM sleep EEG data locked to inhalation peaks, contrasted against randomly selected, matched data segments (sum of t-values across all significant electrodes). The contour lines indicate significant clusters ($p < 0.001$,

corrected), illustrating enhanced power in the SO_spindle range around the inhalation peak (time = 0). The white waveform depicts the inhalation-peak-locked respiration signal (mean \pm SEM across participants). The topography illustrates the statistical results (summed t-values of the cluster) across electrodes (electrode Cz highlighted in black). (b) The modulation index indicates that the phase of respiration significantly influences the EEG amplitude at electrode F2 with local peaks in the SO (0.5 Hz) and spindle range (16 Hz; $z > 1.96$; the red line depicts all significantly modulated frequencies; corrected for multiple comparisons across frequencies). (c) Determining the respiratory phases during the down-states of the detected SOs (top topographical insert, red) and during the onset of spindles (bottom topographical insert, blue) across participants reveals a significant non-uniform circular distribution ($p < 0.05$; Spindles: $p < 0.05$; corrected using FDR⁹). The example circular plot (electrode F2 for SOs and P1 for spindles, highlighted in the topographical insets) illustrates the preferred phases for respiration-SO modulation (red: mean angle = $-12.61^\circ \pm 0.22$, mean vector length = 0.56) and respiration-spindle modulation (blue: mean angle = $29.6^\circ \pm 0.27$, mean vector length = 0.37) in relation to the inhalation peak (i.e., 0°). (d) Determining the respiratory phases during the peak of the detected SO_spindles reveals a significant non-uniform circular distribution ($p < 0.05$; corrected). The example circular plot at electrode F2 (highlighted in the topography) illustrates that the preferred phases of this respiration-SO_spindle modulation peaked right before the inhalation peak (i.e., 0° , mean angle = $-19.89^\circ \pm 0.21$, mean vector length = 0.51). In all participants, SO_spindle rate (relative to SOs) at F2 was non-uniformly distributed across the respiration cycle (all $p < 0.001$; corrected, see right panel). (e) Results of a robust regression exhibiting a significant positive relationship between the respiration – SO_spindle coupling strength (i.e., vector length) and levels of memory reactivation (i.e., decoding accuracies as reported in Schreiner et. al, 2021) at electrode F2 ($F_{1,16} = 5.41$, $p = 0.034$; corrected). Scatter plot illustrates the observed effect.

In addition, we added a dedicated paragraph to the discussion (page 13, line 316). The section reads as follows:

... Another potential caveat is that 17 of our 20 participants were female. It is well known that sex differences affect sleep parameters, sleep related oscillations and in consequence memory consolidation⁶⁹, which might limit the generalizability of our results. Even though removing the data from male participants did not change the main outcomes of our study (see Supplementary Fig. 9), future work will need to address potential gender related differences with regards to the role of respiration in influencing sleep oscillations in detail...

Reviewer 3:

GENERAL

Comment a: In the introduction, respiration is being framed implicitly as a potential explanatory pacemaker for the changes in SO-spindle coupling seen in neurodevelopment and aging. However, the authors do not clarify whether these periods are also accompanied by changes in breathing/respiratory rhythms. This should be explicitly addressed to clarify the hypothesis-driven focus on these interactions.

Response: We thank the Reviewer for this good suggestion. Indeed, the rate of respiration declines from birth to adolescence (Pediatric Advanced Life Support Provider Manual, 2006), reaching adult levels during this period. In addition, sleep related breathing disturbances are common in older adults (McMillan et al., 2016), with the severity of symptoms accelerating with age (Young et al., 2002). Hence, these breathing related changes closely parallel developmental changes in the precision of SO-spindle coupling and might tentatively hint towards an influence of respiration on SO-spindle coupling changes. We added this information to the Introduction (page 3, line 46).

*...These findings beg the question whether there is an additional underlying pacemaker that influences SO-spindle coupling. Respiration has recently been put forward as such a **scaffold** for brain dynamics, as a growing number of findings demonstrate that breathing impacts cognition^{1,11-14} and shapes brain oscillations^{11,15} during wake in humans. **Interestingly, the rate of respiration (i.e., breathing frequency) declines from birth to adolescence¹⁶, while sleep related breathing disturbances are common in older adults¹⁷, with the severity of symptoms accelerating with age¹⁸. Hence, these breathing related changes closely parallel developmental changes in the precision of SO-spindle coupling. However, whether respiration might impact sleep rhythms and ensuing consolidation processes remains unknown...***

Comment b: The term “pacemaker” carries strong causal connotations, and it is not clear that either the study design or the analyses described in the manuscript meet the standard of evidence needed to establish a causal link. While the authors have made considerable effort to demonstrate that the associations themselves are statistically significant beyond chance levels, it is not adequately established that there exists a direct temporal sequence in which inhalation peaks consistently precede sleep oscillations, or indeed that any such sequence may not be explained at least in part by other physiological variables in play. Figure 2d may hint at the temporal precedence of inhalation peaks over max-amplitude spindles, but the very same figure may equally be visually interpreted as showing temporal precedence of max-amplitude SOs over inhalation peaks. The circular distributions suggest that even among spindles, a sizable minority of vectors show maximum amplitude during negative phases of the inhalation (i.e. precede the inhalation peak).

Response: We agree that the term ‘pacemaker’ might have been a too strong word in the context of our study. We hence toned down the language throughout the manuscript. Our data suggest (as visible in Figure 2d) that on average, SOs (locked to the down-states) precede the inhalation peak, while spindles on average closely follow the inhalation peak. However, it is true that with regards to the exact coupling of respiration with sleep related oscillations there is some variability across participants. To get closer to the question whether there is a prototypical cascade of respiration impacting SOs and spindles, we now directly compared the preferred phases of SO and spindle events (Fig. 2d). This approach allows us to assess whether there is a reliable phase shift between these two event types. It has to be noted that our main approach of locking spindle related analyses to the onset of spindles does not favour this question, given that spindles tended to emerge briefly after the inhalation peak, hence in close proximity to SO-down-states (which clustered briefly before the inhalation peak). Hence, we assessed in a first step again the preferred respiration phase for spindles, but this time with regards to spindle peaks. As shown below, similar to spindle onsets, spindle peaks clustered after the inhalation peak (V-test against 0° , mean $V = 5.74 \pm 0.25$, $p < 0.05$, corrected; mean angle of significant electrodes: $50.45^\circ \pm 0.21$, mean vector length = 0.57). Next, we assessed using the circular Watson-Williams test whether there would be a phase shift between SOs and spindles in relation to their modulation by respiration. We found a significant phase shift across a broad cluster (all $p < 0.05$; corrected) indicating that SOs and spindles clustered at different phases of the respiratory cycle, with SOs emerging (on average) before the inhalation peak and spindles following. We think that these results add further evidence to the claim that respiration not only specifically modulates SO and spindle emergence, but achieves this in a well-regulated fashion.

Supplementary Figure 2. (Top) To assess whether there is a reliable phase shift between SOs and spindles in relation to their modulation by respiration we assessed in a first step the preferred respiration phase for spindle coupling with regards to spindle peaks (the original analysis reported in Fig. 2d with regards to spindle onsets impedes such an analysis due to the proximity of spindle onsets to inhalation peaks and hence SO down-states). Similar to spindle onsets, spindle peaks clustered after the inhalation peak (V-test against 0° , mean $V = 5.74 \pm 0.25$, $p < 0.05$, corrected; mean angle of significant electrodes: $50.45^\circ \pm 0.21$, mean vector length = 0.57). (Bottom)

Next, we assessed using the circular Watson-Williams test whether there would be a phase shift between SOs (i.e., downstates) and spindles (i.e., peaks) in relation to their modulation by respiration. We found a significant phase shift across a broad cluster (all $p < 0.05$; corrected) indicating that SOs and spindles clustered at different phases of the respiratory cycle, with SOs emerging before the inhalation peak and spindles following.

Comment c: Although the associative findings are meaningful and insightful, and certainly do not preclude the possibility of causal links, in the absence of either analytical (e.g. Granger causality or transfer entropy), statistical (e.g. structural equation or mediation modeling) or experimental approaches, they fall short of the burden of proof for causal inference in this chicken-and-egg problem. In that respect, it is not clear that the respiratory rhythms are “making” pace, so much as keeping pace with the SOs/SO-spindle complexes. This limitation may merit further discussion, as well as changes to language throughout the manuscript.

Response: The Reviewer is correct that our results are correlational in nature and do not provide evidence for any causal influence of breathing on the timing of sleep oscillations. That said, we now assessed whether respiration drives activity in the SO and sleep spindle range or vice versa using the phase-slope index (PSI, Jiang et al., 2015). The PSI is a metric used to infer the causal direction of oscillatory interactions from electrophysiological time series (Nolte et al., 2010). The cross-frequency PSI was calculated between the respiration signal (filtered around the peak frequency ± 0.05 Hz) and electrode F2 in case of SOs (filtered around 0.3 to 2 Hz) and electrode P1 in case of spindles (filtered around 12 and 18 Hz). In this context, positive values indicate respiration driving SO /sleep spindle activity, while negative values indicate sleep spindles / SOs driving respiration. The obtained data distributions were tested against zero, using paired samples t-tests.

We found that respiration predicted both SO and spindle activity, as evidenced by a positive PSI (SOs: $t_{1,19} = 2.53$; $p = 0.02$; spindles: $t_{1,19} = 2.7$; $p = 0.014$). We added the information to the Supplemental and refer to it in the manuscript (page 6, line 112 for SOs; page 6, line 122 for spindles).

...For results concerning the directional influence of respiration on SOs (indicating that respiration predicted SO activity), see Supplementary Fig. 1...

...For results concerning the directional influence of respiration on spindles (indicating that respiration predicted spindle activity), see Supplementary Fig. 1...

Supplementary Figure 1. Directional influence of respiration on SO and spindle activity. We assessed whether respiration drives activity in the SO and sleep spindle range or vice versa using the phase-slope index (PSI)⁸. The cross-frequency PSI was calculated between the respiration signal (filtered around the peak frequency ± 0.05 Hz) and electrode F2 in case of SOs (filtered around 0.3 to 2 Hz) and electrode P1 in case of spindles (filtered around 12 and 18 Hz). In this context, positive values indicate respiration driving SO /sleep spindle activity, while negative values indicate sleep spindles / SOs driving respiration. The obtained data distributions were tested against zero, using paired samples t-tests. We found that respiration predicted both SO and spindle activity, as evidenced by a positive PSI (SOs: $t_{1,19} = 2.53$; $p = 0.02$; spindles: $t_{1,19} = 2.7$; $p = 0.014$).

Moreover, as suggested by the Reviewer we added a dedicated section to the discussion delineating potential experiments that might support the idea of a causal influence of breathing on sleep oscillations in future studies (page 12, line 287).

...On that note, it deserves mention that our results are correlational in nature. Hence, future studies, directly manipulating breathing behaviour, will be needed to provide causal evidence for the role of breathing in modulating sleep related oscillations. For example, it has been shown that presenting odours during NREM sleep is capable of modifying respiration during sleep, by decreasing inhalation and increasing exhalation volume for up to 6 breaths⁶⁴. One hypothesis would be that such interference (i.e., shallower inhalation due to odour presentation) might mitigate the modulation of sleep related oscillations during the initial presence of odours. Another line of research potentially providing causal evidence could stem from patients suffering from obstructive sleep apnoea (OSA). OSA is characterized by repetitive upper airway collapse resulting in intermittent hypoxia⁶⁵. Strikingly, in OSA patients' sleep seems to have lost its beneficial effect on memory⁶⁶. While it has been shown that abnormal sleep spindle properties are associated with OSA⁶⁷, it remains unclear whether OSA also impacts the emergence of SOs and SO_spindle complexes. This would corroborate the notion that pathological, arrhythmic breathing affects the coordination of sleep oscillations. Importantly, showing that restoring normal breathing behaviour (e.g., via Continuous Positive Airway Pressure (CPAP) treatment) also restores the influence of respiration on sleep oscillations would lend some causal evidence for the role of breathing on the coordination of sleep oscillations...

Comment d: Additionally, the mechanistic and neurobiological justifications for framing respiratory oscillations as pacemakers are not sufficiently detailed. For instance, under the pacemaker framework, it would be important to offer an explanation for how an inhalation may induce neural expression, and especially how any physiological changes tied to an inhalation may propagate with such speed to thalamocortical and cortical expression of neurophysiology.

Response: We added a dedicated section to the discussion outlining how breathing might influence brain activity, covering also the next comment (Comment e) concerning the impact of the brainstem.

The section reads as follows (page 11, line 266).

...How might breathing exert its impact on brain activity? When we inhale, the incoming airflow stimulates mechanoreceptors of the olfactory sensory neurons⁵⁶. These in turn produce breathing-locked oscillations, which are conveyed to the olfactory bulb and further to the olfactory cortex⁵⁷. The olfactory cortex, however, is not the last terminal of breathing-entrained rhythms. The impact of breathing on neuronal activity has been identified in rodent models and humans in various brain areas^{20-22,40,41,43,44,58}. This anatomical circuitry is in line with recent findings indicating a privileged role of nose breathing as compared to mouth breathing in synchronizing neuronal oscillations and affecting memory processes^{17,39}. However, the brainstem houses respiratory rhythm generators, which might likewise account for the breathing- related entrainment of neuronal oscillations^{43,59} (irrespective of nose or mouth breathing). Specifically, the phases of the respiratory cycle, comprising inhalation and exhalation, are governed by brainstem circuits⁵⁹. The brainstem, as major control hub of the autonomous nervous system, likewise impacts the activity of other vital functions such as the heart rate or gastric functions^{60,61}. Cardiac activity and respiration are intimately linked⁶², while cardiac rhythms and in particular heart rate variability have been shown not only to covary with different sleep stages but also with SO and sleep spindle activity during NREM sleep⁶³. Hence, the extent to which sensory (i.e., olfactory bulb route) or non-sensory (i.e., brainstem route) breathing-locked inputs drive neuronal activity and whether they innervate the same target regions in the brain remains unclear. Controlling the breathing route during sleep would permit drawing more causal inferences whether nose breathing is indeed key for clocking neuronal activity during sleep...

Comment e: For the above reasons, the findings should be framed and interpreted primarily as being associative. Moreover, it is possible that it is not respiration per se, but another physiological pacemaker that paces both oscillations and respirations that may be driving both signals downstream. Namely, autonomic output such as heart rate variability are also strongly coupled to breathing, and as such- it is likely that the greater pacemaker in question may actually be the autonomic nervous system itself. Use of the term pacemaker suggests that there is a neurobiological mechanism coupling breathing to sleep oscillations, which this experimental set up is not suited to test. It is equally likely that the ANS is the pacemaker driving respiration and neural dynamics as downstream targets. This should be addressed in some way. In short, the findings have not been appropriately contextualized within a multidimensional framework of interacting, synchronous, and regulated physiological control systems. Cardiac rhythms, and in particular heart rate variability, are known to be associated with both sleep neurophysiology as well as circulatory physiology both during and outside of sleep, and, in the absence of further analyses, they should at the very least be included in any discussion of pacemaking/pacekeeping during sleep. The role of autonomic-central coupling is consequently also an overlooked point of discussion for this study. The potential mechanistic role of the brainstem, which serves as a true pacemaker for the respiratory system as well as

the cardiac system, is not addressed at all within this framework, and also merits addressing at least in the discussion.

Response: We now addressed the potential role of the brainstem and autonomous nervous system in the discussion (please see response above).

COMMENTS SPECIFIC TO RESULTS & METHODS

Comment 1: Figure 2a is interpreted as showing evidence for enhanced SO/spindle power around the inhalation peak, but the time-frequency t-map clearly depicts a broadband and diffuse, seemingly non-specific cluster enclosed by the contour. This should be more adequately addressed, as the interpretation may not be entirely accurate. For example, it is possible that the plot in Figure 2a is influenced by some form of movement-related or other form of artifact during inspiration and expiration. This should be addressed. The authors should also clarify why, if they think the PAC frequencies are truly neural signal-related, the focus should be specifically on slow oscillations and sleep spindles when there is clear coupling across theta and alpha frequencies as well.

Response: The Reviewer is correct that the time-frequency representation in Fig. 2a exhibits inhalation-locked power increases across a broad range of frequencies. However, as elaborated below, the data suggest that effects are indeed driven by SOs and (fast) sleep spindles, with effects in other frequency bands resulting from frequency smearing.

First, we realized that our way of illustrating the obtained effects, i.e., plotting the summed t-values, blurs the fact that the significant cluster comprises two distinct frequency ranges: one spanning slow frequencies (< 10 Hz) and one spanning faster frequencies (14-25 Hz). This becomes more apparent when plotting the of mean of z-values across all significant electrodes instead of the summed t-values (please see below - we updated Fig. 1a accordingly). Importantly, oscillatory power associated with SOs and sleep spindles is usually not solely confined to the classical frequency range of <1 Hz (SOs) and 12-16 Hz (spindles). This might reflect slight variations in centre frequencies across individual events or participants, or simply the spectral smoothing inherent in TFR analyses (for corresponding effects see e.g. Cox et al., 2014; Weber et al., 2021; Schreiner et al., 2021).

The same points as above (oscillatory power associated with SOs and sleep spindles are not confined to their peak frequencies) relates to the results of the Modulation Index. While the Modulation Index exhibits clear peaks in the SO (0.5 Hz) and spindle range (15.5 Hz), the fact that also adjacent low frequencies of the EEG were impacted by respiration (and frequencies > 16 Hz) likely results from frequency smearing). Importantly, we used a complementary approach that uses discrete, algorithmically detected SOs (0.3 – 1.25 Hz) and spindles (12-18 Hz) for determining the preferred phase of the respiration-SO and respiration-spindle coupling, respectively (Figure 2D). The results of these analyses likewise indicate that SOs and sleep

spindle activity culminated just before (in case of SOs) and right after the inhalation peak (in case of spindles), speaking against the notion that the TFR and PCA related outcomes were driven by (respiration locked) movements or any other artifact.

Time-frequency representation of NREM sleep EEG data locked to inhalation peaks, contrasted against randomly selected, matched data segments (mean of z-values across all significant electrodes). The contour lines indicate significant clusters ($p < 0.001$, corrected), illustrating enhanced power in the SO_spindle range around the inhalation peak (time = 0). The white waveform depicts the inhalation-peak-locked respiration signal (mean \pm SEM across participants).

The modulation index indicates that the phase of respiration significantly influences the EEG amplitude at electrode F2 (highlighted in the topographical inset) with local peaks in the SO (0.5 Hz) and spindle range (15.5 Hz; $z > 1.96$; the red line depicts all significantly modulated frequencies; corrected for multiple comparisons across frequencies).

Comment 2: The choice of, and reliance on data from channels Cz in some places and F2 in others is not adequately explained or justified. In the methods subsection entitled “Modulation Index”, it is neither immediately apparent, nor adequately justified why the channel Cz was selected for this analysis. Channel F2 is initially considered as a representative example, but is later the sole channel used for the analyses shown in Figure 3. While the distribution of preferred phase of coupling between respiratory rhythms and SOs, spindles, and SO-spindle complexes have been shown in supplementary figure 1, the higher order analyses of cognition appear to be focused purely on F2. The manuscript should spend more text explicitly justifying the logic behind choosing the electrode derivations within each analysis or stick with a consistent electrode across analyses, to minimize the appearance of potential cherry picking.

Response: We thank the Reviewer for raising this important point. In brief, our conclusions do not hinge upon the particular choice of electrodes. For consistency, we now use F2 as

representative electrode for all SO and SO-spindle related analyses (also including the Modulation Index in Fig. 1c), while P1 is used for illustrating spindle-related analyses (Fig. 2d). For full transparency we now not only show the average of all significant channels for preferred coupling analyses (SOs, spindles and SO_spindles; see Supplementary Fig. 10; also see below), but also display the distributions at the electrodes exhibiting the strongest modulation by respiration (please see second figure below; Supplementary Fig. 11). We added the Figure to the supplemental and refer to it in the Methods section, where we explain the rationale (page 15, line 376).

...We used electrode F2 as representative electrode for all SO and SO_spindle related analyses and electrode P1 in case of spindles. In Supplementary Fig. 10 we highlight the average of all significant electrodes for phase-related analyses and in Supplementary Fig. 11 we show the electrodes exhibiting the strongest phase-modulation related effects...

Supplementary Figure 10. Preferred coupling phases across all significant electrodes. Preferred phases of the coupling between respiration and SOs (red, mean angle: $-9.4^\circ \pm 0.22$), spindles (blue, mean angle: $16.95^\circ \pm 0.23$) and SO_spindle complexes (green, mean angle: $-4.5^\circ \pm 0.16$) averaged across all significant electrodes exhibit similar phase distribution as illustrated in Fig. 2d and Fig. 3a. A significant non-uniform circular distribution was found for SOs, spindles and SO_spindles (V-test against 0° , SOs: $V = 10.11$, $p < 0.001$; Spindles: $V = 9.42$, $p = 0.0014$; SO_spindles: $V = 14.72$, $p < 0.001$).

Supplementary Figure 11. Electrodes exhibiting the strongest preferred coupling. The strongest preferred coupling between SOs and respiration (in terms of V-values) became apparent at electrode F2 (red: mean angle: $-9.4^\circ \pm 0.22$; V-test against 0° , $V = 10.87$, $p < 0.001$). In case of spindles the strongest preferred coupling was located at electrode CP1 (blue: mean angle: $16.95^\circ \pm 0.23$; V-test against 0° , $V = 9.99$, $p < 0.001$). For SO_spindles electrode the strongest modulation by respiration was detectable at electrode C4 (green: mean angle: $-4.5^\circ \pm 0.16$; V-test against 0° , $V = 13.34$, $p < 0.001$).

Furthermore, as suggested by the Reviewer, we assessed whether the likelihood of SOs to group spindles is modulated by the respiration phase across electrodes. We binned the number of SO_spindles relative to all SOs at each electrode across the respiration cycle in 20 evenly spaced bins (i.e., from $-p$ to p). Next, we averaged the distributions per electrode across

participants and assessed (non)uniformity of the resulting distribution using the Kolmogorov-Smirnov test. Non-uniformity peaked at frontal, central and parietal electrodes indicating that the likelihood of SOs to group with spindles was robustly modulated by respiration. We added the information to the Supplemental and refer to it in the manuscript (page 8, line 179).

...indicating a preferential modulation of coupled SO_spindle events by respiration phase (see Supplementary Fig. 5 for results across electrodes...

Supplementary Figure 5. Modulation of the likelihood of SOs to group spindles by the respiration phase across electrodes. The number of SO_spindles relative to all SOs at each electrode was binned across the respiration cycle in 20 evenly spaced bins (i.e., from $-\pi$ to π in steps of 0.31 radians). Next, the distributions were averaged per electrode across participants and (non)uniformity of the resulting distribution was assessed using the Kolmogorov-Smirnov test. Non-uniformity peaked at frontal, central and parietal electrodes indicating that the likelihood of SOs to group with spindles was robustly modulated by respiration.

Comment 3: In the topographical plots in Figure 3a and 3b, aside from channel F2, there appears to be a lack of agreement in the regional effects shown. The other 2 channels shown in Figure 3b appear to coincide with regions of null effects in Figure 3a, and this is not adequately explained. In fact, it seems that if viewed superimposed, the two topoplots appear to be approximately inverted. This should be addressed in the interpretation and discussion.

Response: In the course of the revision we found a slight mistake in the illustration of the SO_spindle – respiration modulation (Fig. 3a). We realized that the figure was erroneously based on the non-collapsed 40 datasets of our 20 participants. We corrected this mistake, by collapsing across both sessions per participant before statistical assessment and plotting, as it was done in any of the other analyses (please note that only the illustration of the topography was affected and no major differences became apparent; please see below). We updated Fig. 3a and apologize for the mistake.

That said, the Reviewer is correct that one electrode (FC2) of the significant cluster in Fig. 3b (coupling-decoding correlation) did not show a significant non-uniform distribution with regards to SO_spindle – respiration coupling (Fig. 3a). However, even though the distribution at FC2 did not reach significance ($V = 5.11$, $p = 0.052$), SO-spindles tended to cluster towards the inhalation peak (mean direction: $1.4 \pm 0.21^\circ$; vector length: 0.25), indicating that results at electrode FC2 were not qualitatively different.

-- SO_spindle - respiration coupling [FC2] --

Comment 4: Given the focus on respiratory oscillations, the authors do not address whether their cohort was screened for sleep-disordered breathing (SDB, e.g., obstructive sleep apnea). Although the PSQI was used to assess sleep quality in this otherwise healthy group, this instrument is not as well-suited nor as sensitive to SDB as other, more targeted questionnaires such as STOP-BANG or the Berlin Questionnaire. Furthermore, other measurements associated with incidence of SDB, such as body-mass index, neck circumference, or waist-to-hip circumference ratio were neither reported nor modeled as analytical covariates. SDB is known to cause sleep fragmentation, due to which oscillatory expression tends to localize to periods without cessation in breathing, which in turn is associated with changes in SO-spindle coupling in individuals with SDB. Given the multifarious links between SDB and altered neural and respiratory rhythms, as well as associated cardiovascular and cerebrovascular impacts, this is an important limitation to address in the discussion. It is possible that breathing abnormalities in the signal may alter SO-spindle or SO-spindle-respiratory coupling in a way that biased the current analyses. This should be addressed in some form.

Response: The Reviewer is right that we did not specifically assess sleep disordered breathing (SDB) in our participants, which might have considerably impacted the results of our work. However, SDB is well known to cause sleep fragmentation (e.g., Kimoff et al., 1996). All of our participants exhibited healthy sleep and no signs of sleep fragmentation (i.e., elevated number of awakenings as assessed with sleep scoring). Moreover, we carefully inspected the respiratory data during recording and offline pre-processing and found no signs of breathing cessation. Finally, sleep oscillations and specifically sleep spindle properties (i.e., frequency and topography) have been shown to be altered by SDB (e.g., Weiner et al., 2016). Spindle characteristics were generally within the expected range, both in terms of frequency and topography. Hence, it seems rather unlikely that SDB had an influence on the reported results, despite not being explicitly assessed.

We added this information to the discussion. The section reads as follows (page 12, line 305):

...It has to be noted that we did not explicitly screen for sleep disordered breathing (SDB) conditions in our participants, which might have considerably impacted the results of our work. However, SDB is well known to cause sleep fragmentation (e.g.,⁶⁸). All our participants exhibited healthy sleep and no signs of sleep fragmentation (i.e., elevated number of awakenings) as assessed with sleep scoring. Moreover, we carefully inspected the respiratory data during recording and offline pre-processing and found no signs of breathing cessation. Finally, sleep oscillations and specifically sleep spindle properties (i.e., frequency and topography) have been shown to be altered by SDB (e.g.,⁶⁷). Spindle characteristics were generally within the expected range, both in terms of frequency and topography. Hence, it seems rather unlikely that SDB had an influence on the reported results, despite not being explicitly assessed...

Comment 5: When describing the memory reactivation results associated with Figure 3b, the authors refer to a prior publication to describe the LDA decoding technique. However, in the interest of clarity, at least a concise (1-2 sentence) elaboration on this approach should be included in the methods section of the current manuscript considering it is a central part of the analysis.

Response: We added a paragraph termed *Multivariate analysis* to the Methods section explaining the rationale of the decoding procedure (please see page 18, line 484).

...Multivariate analysis (brief description, for details see⁴). Multivariate classification of single-trial EEG data was performed using MVPA-Light, a MATLAB-based toolbox for multivariate pattern analysis¹⁹. For all multivariate analyses, a LDA was used as a classifier¹⁹. Prior to classification, all data were re-referenced using a common average reference (CAR).

To investigate differential evidence for object vs. scene representations as a function of prior learning during SO-spindle complexes, we used the temporal generalization method²⁰. Prior to decoding, a baseline correction was applied based on the whole trial ([-0.5 to 3 s] for localizer segments; [-1.5 to 1.5 s] for SO-spindle segments). Next, localizer and sleep data were z-scored across trials and collapsed across sessions. PCA was applied to the pooled wake-sleep data and the first 30 principal components were retained. Localizer and sleep data were smoothed using a running average window of 150 ms. A classifier was then trained for every time point in the localizer data and applied on every time point during SO-spindle complexes. No cross-validation was required since localizer and sleep datasets were independent. As metric, we used the area under the curve. For statistical evaluation, surrogate decoding performance was calculated by shuffling the training labels (stemming from the localizer task) 250 times. Again, the resulting performance values were averaged, providing baseline values for each participant under the null hypothesis of label exchangeability...

MINOR

Comment 1: "SO_spindles" and "SO_spindle complexes" should be replaced with SO-spindles/SO-spindle complexes as needed.

Response: We introduced the term "SO_spindles" as we felt that terms like

“SO-spindle – respiration modulation” would be hard to read. That is the reason why we would prefer sticking with the current terminology. If the Reviewer still thinks that “SO-spindles” would be preferable we are happy to change the wording after the revision.

Comment 2: In the introduction, there is a reference to “in- and decreases in memory...” which should be changed to increases and decreases.

Response: We rephrased the sentence.

*... Moreover, its precision increases from childhood to adolescence²¹ and then declines again during ageing^{22,23}, with concomitant **increases** and decreases in memory performance...*

Comment 3: In the methods, in the subsection entitled “Experimental Overview”, there appears to be a typographical error – the study start time is indicated as being 12am, but the nap initiation occurred at 1pm. If inaccurate, the start time of midnight should be rectified to noon.

Response: We changed the wording (page 14, line 356).

*...At around 12 **p.m.** the experiment started with a modified version of the psychomotor vigilance task (“PVT”²⁴)...*

Comment 4. In the methods subsection entitled “Modulation Index”, “z-transformed” should be replaced with z-scored/z-normalized (as used elsewhere in the manuscript), mainly to avoid confusion with the z-transform, which is an altogether different concept in signal processing.

Response: We changed the wording to (page 17, line 442). The sentence now reads as follows:

*...The mean and standard deviation across the reference distribution was then used to **z-score** the MI of the empirical data...*

Comment 5. In the methods subsection entitled “SO-spindle rate across respiratory cycle”, there is a typographical error – “-pi to pi”. Further, since this range is in radians, the step size of 18 degrees should also be converted to and reported in radians.

Response: We corrected the mistake and converted the degrees to radians (page 18, line 478).

*... Next, we binned the number of detected SO_spindles at electrode F2 across the respiration cycle in 20 evenly spaced bins (i.e., from **-pi** to **pi** in steps of **0.31 radians**)...*

References:

- Schreiner, T., Petzka, M., Staudigl, T. & Staresina, B. P. Endogenous memory reactivation during sleep in humans is clocked by slow oscillation-spindle complexes. *Nat Commun* **12**, 3112 (2021).
- Ngo, H.-V. V. & Staresina, B. P. Shaping overnight consolidation via slow-oscillation closed-loop targeted memory reactivation. *Proc Natl Acad Sci U S A* **119**, e2123428119 (2022).
- Hahn, M. A., Heib, D., Schabus, M., Hoedlmoser, K. & Helfrich, R. F. Slow oscillation-spindle coupling predicts enhanced memory formation from childhood to adolescence. *Elife* **9**, e53730 (2020).
- Muehlroth, B. E. *et al.* Precise Slow Oscillation-Spindle Coupling Promotes Memory Consolidation in Younger and Older Adults. *Sci Rep* **9**, 1940 (2019).
- Cox, R., Driel, J. van, Boer, M. de & Talamini, L. M. Slow Oscillations during Sleep Coordinate Interregional Communication in Cortical Networks. *J. Neurosci.* **34**, 16890–16901 (2014).
- Weber, F. D. *et al.* Coupling of gamma band activity to sleep spindle oscillations - a combined EEG/MEG study. *Neuroimage* **224**, 117452 (2021).
- Mölle, M., Bergmann, T. O., Marshall, L. & Born, J. Fast and Slow Spindles during the Sleep Slow Oscillation: Disparate Coalescence and Engagement in Memory Processing. *Sleep* **34**, 1411–1421 (2011).
- Jiang, H., Bahramisharif, A., van Gerven, M. A. J. & Jensen, O. Measuring directionality between neuronal oscillations of different frequencies. *Neuroimage* **118**, 359–367 (2015).
- Nolte, G., Ziehe, A., Krämer, N., Popescu, F. & Müller, K. Comparison of Granger Causality and Phase Slope Index. in (2008).
- Weiner, O. M. & Dang-Vu, T. T. Spindle Oscillations in Sleep Disorders: A Systematic Review. *Neural Plast* **2016**, 7328725 (2016).
- Fitzpatrick, M. F. *et al.* Effect of nasal or oral breathing route on upper airway resistance during sleep. *Eur Respir J* **22**, 827–832 (2003).
- McMillan, A. & Morrell, M. J. Sleep disordered breathing at the extremes of age: the elderly. *Breathe (Sheff)* **12**, 50–60 (2016).
- Young, T., Peppard, P. E. & Gottlieb, D. J. Epidemiology of obstructive sleep apnea: a population health perspective. *Am J Respir Crit Care Med* **165**, 1217–1239 (2002).

REVIEWER COMMENTS

Reviewer #1 (Remarks to the Author):

I want to thank the authors for their careful and thorough revision. Most of my initial comments have been answered very clearly and only issues remains, which concerns the interpretation of SO_spindle results. Taken together, both the very low number of SO_spindle trials (as evident from the new table) and the fact that FC2 was not included in the cluster of significant electrodes for SO_spindle coupling call for a more conservative phrasing of SO_spindle results. These findings should really not be overstated and I would ask the authors to tone down the Results and Discussion sections in this regard. I regret to say that I do not find the 'qualitative' similarity the authors show with the (uncorrected) analysis of FC2 to be overly convincing, given that i) essentially all neighbouring electrodes form the large cluster and ii) the 'tendency towards peak inspiration' is not trivial to assess (as the distribution once again seems to be multimodal).

As a final note, the authors have decided to largely amend the original manuscript in the Supplement as opposed to making changes in the main text. For illustrative changes like the bar plot vs boxplot question (Suppl. Fig. 8), I would suggest to either remove the suggested new visualisation from the Supplement (and oppose my point, which is perfectly fine) or change the figure in the main text. By no means should new items be added just for the sake of complying with the reviewer's opinion. An unrelated (but stylistic) suggestion would be to use different colour schemes for the topographies of phase shifts in contrast to statistical results (F-/V-values).

Reviewer #2 (Remarks to the Author):

The authors have provided a very careful revision satisfactorily addressing all of my points. I suggest 2 additional minor changes before final acceptance: (i) Please, move Suppl. Fig. 6 regarding the link between respiration and SO-spindle phase-coupling to the main text. This data is more important than SO-spindle co-occurrence and, in my view nicely illustrates that in addition to the maximum at the inhalation peak also a subsequent minimum during expiration might contribute to the nonuniform phase distribution. (ii) Though negative, findings (plus figure) regarding my Comment #3 should be added to the supplementary materials.

Reviewer #3 (Remarks to the Author):

The authors have adequately addressed most of our concerns from the prior review in the current revised manuscript. However, there are some concerns, mostly minor, that remain. We think these concerns could be addressed in a minor revision. Our remaining concerns are outlined below.

As a general note, we think the inclusion of the PSI results add strength to the manuscript and the authors could consider moving them to the main text from the supplements to more strongly support their inferences regarding modulation of slow oscillation and spindle expression instead of just co-occurrence.

1. Phase locking of spectral power to inhalation peak: The authors note that the prior plotting of summed t-values obscured distinct clusters in frequency space, and now plot the mean z-values across significant electrodes. It does appear apparent that there are two distinct peaks, as the authors note, one between <1Hz and 10Hz and one between 14-25Hz. The authors then note that the blurring is likely due to frequency smearing. While we agree that frequency smearing is likely to occur in these kinds of analyses, the frequency range of the effect is quite large for both clusters and it seems unclear how the results could be confidently interpreted as definitively localized to the slow oscillation

(0.3-1.25Hz) frequency range and the spindle frequency range (~11-~16Hz). There could be multiple possible reasons for the large frequency range in the summed t values and the mean z-values representations. One, it could be that the frequency specificity of phase locking of spectral power to inhalation peaks may vary widely across individuals, with some phase locking their low frequency closer to 10 hz than 1 Hz and others closer to 1 Hz than 10 hz, i.e., there may be a distribution across subjects in peak frequency of phase locking. The same may be true for the higher frequency cluster. To address this possibility, we suggest showing plots for a subset of representative subjects to illustrate the range of individual phenotypic expression of EEG-respiratory coupling, and to provide valuable insight into the underlying individual differences in sleep physiology as it relates to respiration. The frequency of peak phase locking may be relevant for individual differences in sleep-related cognitive or health outcomes and it would be good to know if there is considerable variability in the peak frequencies of these relationships. Another possibility is that these clusters represent more than two clusters but the degree of frequency smearing obscures the true nature of coupling across frequency space. This could be consistent with the work of Brice McConnel (doi: 10.1093/sleep/zsab125) which shows clear distinct coupling of theta and multiple sigma clusters, some of which stretch into the beta frequency range, with slow oscillations. It is therefore possible that slow oscillation and theta clusters could be smeared into one while all the sigma clusters could be smeared into one. This may also explain the degree of frequency smearing. Showing plots for a subset of representative subjects in the supplements may also address this possibility, because it could show examples, at least for some subjects, of multiple isolated peaks that are consistent across subjects versus two peaks that vary across frequency space from individual to individual. If the authors still think these two possibilities are unlikely and that the frequency ranges of these effects are driven solely by frequency smearing in the results, we think there are analytic approaches that could be implemented that could demonstrate the relative impact of frequency smearing on the results. These should be shown in supplements to bolster this argument. Regardless, due to the degree of frequency smearing, it is difficult to say with confidence that the phase locking of spectral power to inhalation peaks are definitively due to the phase locking of solely slow oscillations and sleep spindles from the reported results and this should be addressed in text revisions associated with this interpretation.

2. We thank the authors for their attempt to address the concerns raised regarding sleep disordered breathing as a possible confound. However, we think the approaches taken do not adequately justify the strength of the interpretation given that it is "rather unlikely that SDB had an influence on the reported results, despite not being explicitly assessed". In short, we do think the authors have responded appropriately by acknowledging the limitation that their cohort was not prescreened or evaluated for SDB. However, we feel that categorical clarity is needed in the discussion of this limitation. The authors indicate that they visually inspected respiratory signals and assert that no breathing cessation occurred and infer by proxy that since sleep neurophysiology as observed in their dataset appears to conform to normative standards, that sleep disordered breathing is unlikely to be impacting their data. However, as the study protocol did not feature collection of pulse oximetry or plethysmography, and as the authors have not reported evaluation of the data by clinicians, it is difficult to accept their assertion at face value. Indeed, the authors assert that they saw no signs of sleep fragmentation and no signs of breathing cessation in any of their subjects. Given the average AHI and level of sleep fragmentation across the lifespan in non-clinical cohorts drawn from community samples, we think this is unlikely to be true if the subjects were assessed with gold standard clinical polysomnography, scored by trained clinicians board certified in sleep medicine. Even a thorough evaluation of breathing cessation based on respiratory belts alone may fail to capture instances of hypoxemia, and therefore it is not possible to definitively preclude the possibility of SDB influencing the findings. We also think, without clinical training, it can be difficult to identify true arousals in PSG data. This should at least be more adequately acknowledged as a limitation in the discussion. We do not think this invalidates the important findings described in the revised manuscript, but we do think this limitation should not be dismissed either.

Minor comments:

1. Some of the revised text includes typographical errors. One example is the caption under supplementary fig 5, which should read pi to pi and not -p to pi and the parentheses should be closed out.

2. We think that SO_Spindle as a term in printed text is not typical of standard formatting. However, we think this is a decision best decided by the editors and the authors and not us as reviewers.

REVIEWER COMMENTS

Reviewer #1

Comment 1: I want to thank the authors for their careful and thorough revision. Most of my initial comments have been answered very clearly and only issues remains, which concerns the interpretation of SO_spindle results. Taken together, both the very low number of SO_spindle trials (as evident from the new table) and the fact that FC2 was not included in the cluster of significant electrodes for SO_spindle coupling call for a more conservative phrasing of SO_spindle results. These findings should really not be overstated and I would ask the authors to tone down the Results and Discussion sections in this regard. I regret to say that I do not find the 'qualitative' similarity the authors show with the (uncorrected) analysis of FC2 to be overly convincing, given that i) essentially all neighbouring electrodes form the large cluster and ii) the 'tendency towards peak inspiration' is not trivial to assess (as the distribution once again seems to be multimodal).

Response: We take the Reviewer's point and toned down the phrasing with regard to the SO_spindle results in the results and discussion sections (changes are marked in the main text).

As a final note, the authors have decided to largely amend the original manuscript in the Supplement as opposed to making changes in the main text. For illustrative changes like the bar plot vs boxplot question (Suppl. Fig. 8), I would suggest to either remove the suggested new visualisation from the Supplement (and oppose my point, which is perfectly fine) or change the figure in the main text. By no means should new items be added just for the sake of complying with the reviewer's opinion. An unrelated (but stylistic) suggestion would be to use different colour schemes for the topographies of phase shifts in contrast to statistical results (F-/V-values).

Response: In light of the Reviewer's suggestion, we removed Supplementary Fig. 8. In addition, we now use a different colour scheme for topographies depicting phase values (see Supplementary Fig. 4; we kept the old colour scheme for all statistic-related topographies).

Supplementary Figure 4. Topographies of preferred phases for SO-spindle modulation by respiration (left: based on circular mean; right: based on circular median). The topographies illustrate that SO_spindles tended to occur briefly before the inhalation peak (i.e., 0°) across the cluster where significant phase modulation was detectable (as shown in Fig. 3a).

Reviewer #2

Comment 1: The authors have provided a very careful revision satisfactorily addressing all of my points. I suggest 2 additional minor changes before final acceptance: (i) Please, move Suppl. Fig. 6 regarding the link between respiration and SO-spindle phase-coupling to the main text. This data is more important than SO-spindle co-occurrence and, in my view nicely illustrates that in addition to the maximum at the inhalation peak also a subsequent minimum during expiration might contribute to the nonuniform phase distribution. (ii) Though negative, findings (plus figure) regarding my Comment #3 should be added to the supplementary materials.

Response: As suggested by the Reviewer we added the analysis concerning the association between respiration – SO coupling strength and levels of memory reactivation, as well as between respiration – spindle coupling strength and levels of memory reactivation to the Supplement and refer to it in the manuscript (page 9, line 202).

...($p < 0.05$, corrected for multiple comparisons across electrodes using FDR³⁰; for the relationship between respiration-SO_spindle coupling strength and behavioural levels of memory consolidation see Supplementary Fig. 6; *conducting robust regressions between levels of memory reactivation and respiration-SO coupling or respiration-spindle coupling independently did not yield any significant effect, see Supplementary Fig. 7*)...

Supplementary Figure 7. Coupling strength [SO-respiration, spindle-respiration] and memory reactivation. Results of a robust regression exhibiting no significant positive relationship between the respiration – SO coupling strength (i.e., vector length) and levels of memory reactivation (all $p > 0.2$) and between respiration – spindle coupling strength and levels of memory reactivation (all $p > 0.1$).

We also agree that Supplementary Fig. 6, illustrating the consistency of SO_spindle coupling across respiration is of potential interest. Nevertheless, we would prefer to keep it in the Supplement. This result is based on the computation of the mean resultant vector length for SO-spindle coupling for each of the 20 phase bins across the respiratory cycle. This means that the analysis was based on a modest number of (SO-spindle) trials per phase bin, which might affect the stability of the procedure. While we do not think that this fact invalidates the result, we would prefer not putting it in the main text. However, we believe that this, as of now, preliminary result might inspire future work making use of full night recordings, thereby resorting on a higher number of SO_spindle events. We added the information that the result illustrated in Supplementary Fig. 6 should be considered preliminary to the figure legend (please see below).

Supplementary Figure 6. To assess whether the consistency of SO-spindle coupling is modulated by the respiration phase, we determined in each participant the respiratory peak frequency and filtered the respiratory data (locked to inhalation peaks) around the peak frequency (± 0.05 Hz, two-pass Butterworth bandpass filter, order = three cycles of the low frequency cut-off). Then a Hilbert transform was applied, and the instantaneous phase angle was extracted. Next, we binned the data across the respiration cycle in 20 evenly spaced bins and computed the vector length, hence the amount of SO-spindle phase-amplitude coupling at electrode F2. (Non)uniformity of the resulting distribution was assessed per participant using the Kolmogorov-Smirnov test. We found non-uniform distributions in all participants (all $p < 0.01$; corrected for multiple comparisons across participants using FDR correction), indicating that SO-spindle coupling was moderately impacted by the respiratory phase with the highest vector length around the inhalation peak (i.e., 0° ; bars reflect the average vector length \pm SEM per bin). **It has to be noted that this analysis is based on a modest number of SO_spindles per phase bin, rendering the results preliminary. Future work, making use of whole night recordings, will need to assess their stability on the basis of a higher number of SO_spindle events.**

Reviewer #3

The authors have adequately addressed most of our concerns from the prior review in the current revised manuscript. However, there are some concerns, mostly minor, that remain. We think these concerns could be addressed in a minor revision. Our remaining concerns are outlined below.

As a general note, we think the inclusion of the PSI results add strength to the manuscript and the authors could consider moving them to the main text from the supplements to more strongly support their inferences regarding modulation of slow oscillation and spindle expression instead of just co-occurrence.

Response: We agree with the Reviewer and moved the PSI results to the main text and Figure 2.

Page 6 line 112:

... Finally, we quantified the directional influence of respiration on SO activity at electrode F2 using the Phase Slope Index (PSI³¹). We found that respiration predicted SO activity, as evidenced by a positive PSI (mean PSI: 0.0003 ± 0.0001 ; t- test against zero: $t_{1,19} = 2.17$; $p = 0.042$; see Figure 2e) ...

Page 6 line 123:

... Again, we quantified the directional influence of respiration on spindle activity at electrode P1 using the PSI³¹. As with SOs, we found that respiration predicted spindle activity, as evidenced by a positive PSI (mean PSI: 0.0024 ± 0.0007 ; t- test against zero: $t_{1,19} = 3.1$; $p = 0.005$; see Figure 2e) ...

Fig. 2 Respiration shapes sleep rhythms. (a) Time–frequency representation of NREM sleep EEG data locked to inhalation peaks, contrasted against randomly selected, matched data segments (mean of z-values across all significant electrodes). The contour lines indicate significant clusters ($p < 0.001$, corrected), illustrating enhanced power in the SO_spindle range around the inhalation peak (time = 0). The white waveform depicts the inhalation-peak-locked respiration signal (mean \pm SEM across participants). The topography illustrates the statistical results (summed t-values of the cluster) across electrodes. (b) Source-space data suggest that TFR results emerge from fronto-parietal areas and the right medial temporal lobe ($p = 0.037$, corrected). (c) The modulation index indicates that the phase of respiration significantly influences the EEG amplitude at electrode F2 (highlighted in the topographical inset) with local peaks in the SO (0.5 Hz) and spindle range (15.5 Hz; $z > 1.96$; the red line depicts all significantly modulated frequencies; corrected for multiple comparisons across frequencies). (d) Determining the respiratory phases during the down-states of the detected SOs (top topographical insert, red) and during the onset of spindles (bottom topographical insert, blue) across participants reveals a significant non-uniform circular distribution (SOs: mean $V = 7.42 \pm 0.28$, $p < 0.05$; Spindles: mean $V = 6.84 \pm 0.24$, $p < 0.05$; corrected using FDR³⁰). The example circular plot (electrode F2 for SOs and P1 for spindles, highlighted in the topographical insets) illustrates the preferred phases for respiration-SO modulation (red: mean angle = $-21.36^\circ \pm 0.20$, mean vector length = 0.58) and respiration-spindle modulation (blue: mean angle = $14.06^\circ \pm 0.23$, mean vector length = 0.44) in relation to the inhalation peak (i.e., 0°). The right panel illustrates the temporal modulation of SOs (red) and spindles (blue; incidence in percent of events detected in time window of interest (i.e., inhalation peak ± 1.5 s) by respiration at electrodes F2 (SOs) and P1 (spindles). Solid red (SOs) and blue (spindles) horizontal lines indicate significant differences, resulting from comparison with event-free control events ($p < 0.05$; corrected). (e) Directional respiration-SO (top) and respiration-spindle coupling (bottom), as obtained by the phase-slope index (PSI; mean \pm SEM). Respiration-phases significantly predicted both SO (t-test against zero, two-sided: $t_{1,19} = 2.17$; $p = 0.042$) and spindle amplitudes (t-test against zero, two-sided: $t_{1,19} = 3.1$; $p = 0.005$).

Comment 1. Phase locking of spectral power to inhalation peak: The authors note that the prior plotting of summed t-values obscured distinct clusters in frequency space, and now plot the mean z-values across significant electrodes. It does appear apparent that there are two distinct peaks, as the authors note, one between $<1\text{Hz}$ and 10Hz and one between $14\text{--}25\text{Hz}$. The authors then note that the blurring is likely due to frequency smearing. While we agree that frequency smearing is likely to occur in these kinds of analyses, the frequency range of the effect is quite large for both clusters and it seems unclear how the results could be confidently interpreted as definitively localized to the slow oscillation ($0.3\text{--}1.25\text{Hz}$) frequency range and the spindle frequency range ($\sim 11\text{--}16\text{Hz}$). There could be multiple possible reasons for the large frequency range in the summed t values and the mean z-values representations. One, it could be that the frequency specificity of phase locking of spectral power to inhalation peaks may vary widely across individuals, with some phase locking their low frequency closer to 10 Hz than 1 Hz and others closer to 1 Hz than 10 Hz , i.e., there may be a distribution across subjects in peak frequency of phase locking. The same may be true for the higher frequency cluster. To address this possibility, we suggest showing plots for a subset of representative subjects to illustrate the range of individual phenotypic expression of EEG-respiratory coupling, and to provide valuable insight into the underlying individual differences in sleep physiology as it relates to respiration. The frequency of peak phase locking may be relevant for individual

differences in sleep-related cognitive or health outcomes and it would be good to know if there is considerable variability in the peak frequencies of these relationships. Another possibility is that these clusters represent more than two clusters but the degree of frequency smearing obscures the true nature of coupling across frequency space. This could be consistent with the work of Brice McConnell (doi: 10.1093/sleep/zsab125) which shows clear distinct coupling of theta and multiple sigma clusters, some of which stretch into the beta frequency range, with slow oscillations. It is therefore possible that slow oscillation and theta clusters could be smeared into one while all the sigma clusters could be smeared into one. This may also explain the degree of frequency smearing. Showing plots for a subset of representative subjects in the supplements may also address this possibility, because it could show examples, at least for some subjects, of multiple isolated peaks that are consistent across subjects versus two peaks that vary across frequency space from individual to individual. If the authors still think these two possibilities are unlikely and that the frequency ranges of these effects are driven solely by frequency smearing in the results, we think there are analytic approaches that could be implemented that could demonstrate the relative impact of frequency smearing on the results. These should be shown in supplements to bolster this argument. Regardless, due to the degree of frequency smearing, it is difficult to say with confidence that the phase locking of spectral power to inhalation peaks are definitively due to the phase locking of solely slow oscillations and sleep spindles from the reported results and this should be addressed in text revisions associated with this interpretation.

Response: The Reviewer makes an excellent point and as suggested, we inspected individual time-frequency representations of five representative participants in order to elucidate the source of the broad frequency clusters. As shown below, these examples speak for the second scenario suggested by the Reviewer. In all five time-frequency representations we found, in addition to clear peaks in the SO and spindle range, distinct peaks in the theta range (albeit with varying peak frequencies). Theta bursts have been reported to be associated with SO down-states (McConnel et al., 2021; Jiang et al., 2019, Gonzales et al., 2018). Hence, the presence of SO-downstate related theta in conjunction with enhanced SO power seems indeed to drive the degree of ‘frequency smearing’ in the low frequencies. For spindles, we did not find any evidence that different spindle clusters might have contributed to our results. Instead, power enhancement in the spindle range was consistently visible between 13 and 20 Hz, peaking towards higher spindle frequencies (i.e., 17-18 Hz; which would correspond to early fast spindles (in relation to SO down-states) as reported in McConnell et al., 2021). Hence, it seems likely that in the case of spindles, slight variations in centre frequencies across individual events (within participants) in combination with the spectral smoothing inherent in TFR analyses underlies the relatively broad spindle cluster in Figure 2a. We added the figure shown below to the Supplement (Supplementary Fig. 1) and refer to it in text (page 5, line 94).

... Respiration-locked time frequency representations (TFRs) exhibited increased power in the SO_spindle range around the inhalation peak, i.e., an initial low frequency burst (comprising peaks in the SO and theta range²⁸⁻³⁰) followed by a fast spindle burst (12-18 Hz; $p < 0.001$, corrected for multiple comparisons across time, frequency, and electrodes; see Fig. 2a; for individual time-frequency representations of five representative participants see Supplementary Fig. 1)...

Supplementary Figure 1. Individual time frequency representations of NREM sleep EEG data locked to inhalation peaks for five representative participants. In addition to clear peaks in the SO and spindle range, distinct peaks in the theta range (albeit with varying peak frequencies between 5 and 8 Hz) became apparent (the black line on the left represents power across time in the significant cluster indicated by the contour line). The presence of NREM theta bursts in conjunction with enhanced SO power putatively underlies the broad low frequency cluster in Figure 2a. Power enhancement in the spindle range was consistently visible across participants between 13 and 20 Hz, peaking towards higher spindle frequencies (i.e., 17-18 Hz).

In addition, we updated Fig. 2a and illustrate power across frequencies (as in the Supplementary Figure above), showing that the low frequency cluster included distinct peaks in the SO and theta range. We also refer to the additional result in the main text (page 5, line 92).

... Respiration-locked time frequency representations (TFRs) exhibited increased power in the SO_spindle range around the inhalation peak, i.e., an initial low frequency burst (including distinct peaks in the SO and theta range²⁸⁻³⁰) followed by a fast spindle burst (12-18 Hz; $p < 0.001$, corrected for multiple comparisons across time, frequency, and electrodes; see Fig. 2a; for individual time-frequency representations of five representative participants see Supplementary Fig. 1)...

Fig. 2 Respiration shapes sleep rhythms. (a) Time–frequency representation of NREM sleep EEG data locked to inhalation peaks, contrasted against randomly selected, matched data segments (mean of z-values across all significant electrodes). The contour lines indicate significant clusters ($p < 0.001$, corrected), illustrating enhanced power in the SO_spindle range around the inhalation peak (time = 0). **The black line on the left illustrates mean z-power averaged across time in the significant cluster (\pm SEM across participants).** The white waveform depicts the inhalation-peak-locked respiration signal (mean \pm SEM across participants). The topography illustrates the statistical results (summed t-values of the cluster) across electrodes. (b) Source-space data suggest that TFR results emerge from fronto-parietal areas and the right medial temporal lobe ($p = 0.037$, corrected). (c) The modulation index indicates that the phase of respiration significantly influences the EEG amplitude at electrode F2 (highlighted in the topographical inset) with local peaks in the SO (0.5 Hz) and spindle range (15.5 Hz; $z > 1.96$; the red line depicts all significantly modulated frequencies; corrected for multiple comparisons across frequencies). (d) Determining the respiratory phases during the down-states of the detected SOs (top topographical insert, red) and during the onset of spindles (bottom topographical insert, blue) across participants reveals a significant non-uniform circular distribution (SOs: mean $V = 7.42 \pm 0.28$, $p < 0.05$; Spindles: mean $V = 6.84 \pm 0.24$, $p < 0.05$; corrected using FDR³³). The example circular plot (electrode F2 for SOs and P1 for spindles, highlighted in the topographical insets) illustrates the preferred phases for respiration-SO modulation (red: mean angle = $-21.36^\circ \pm 0.20$, mean vector length = 0.58) and respiration-spindle modulation (blue: mean angle = $14.06^\circ \pm 0.23$, mean vector length = 0.44) in relation to the inhalation peak (i.e., 0°). The right panel illustrates the temporal modulation of SOs (red) and spindles (blue; incidence in percent of events detected in time window of interest (i.e., inhalation peak ± 1.5 s) by respiration at electrodes F2 (SOs) and P1 (spindles). Solid red (SOs) and blue (spindles) horizontal lines indicate significant differences, resulting from comparison with event-free control events ($p < 0.05$; corrected). (e) Directional respiration-SO (top) and respiration-spindle coupling (bottom), as obtained by the phase-slope index (PSI; mean \pm SEM). Respiration-phases significantly predicted both SO (t-test against zero, two-sided: $t_{1,19} = 2.17$; $p = 0.042$) and spindle amplitudes (t-test against zero, two-sided: $t_{1,19} = 3.1$; $p = 0.005$).

Comment 2. We thank the authors for their attempt to address the concerns raised regarding sleep disordered breathing as a possible confound. However, we think the approaches taken do not adequately justify the strength of the interpretation given that it is “rather unlikely that SDB had an influence on the reported results, despite not being explicitly assessed”. In short, we do think the authors have responded appropriately by acknowledging the limitation that their cohort was not prescreened or evaluated for SDB. However, we feel that categorical clarity is needed in the discussion of this limitation. The authors indicate that they visually inspected respiratory signals and assert that no breathing cessation occurred and infer by proxy that since sleep neurophysiology as observed in their dataset appears to conform to normative standards, that sleep disordered breathing is unlikely to be impacting their data. However, as the study protocol did not feature collection of pulse oximetry or plethysmography, and as the authors have not reported evaluation of the data by clinicians, it

is difficult to accept their assertion at face value. Indeed, the authors assert that they saw no signs of sleep fragmentation and no signs of breathing cessation in any of their subjects. Given the average AHI and level of sleep fragmentation across the lifespan in non-clinical cohorts drawn from community samples, we think this is unlikely to be true if the subjects were assessed with gold standard clinical polysomnography, scored by trained clinicians board certified in sleep medicine. Even a thorough evaluation of breathing cessation based on respiratory belts alone may fail to capture instances of hypoxemia, and therefore it is not possible to definitively preclude the possibility of SDB influencing the findings. We also think, without clinical training, it can be difficult to identify true arousals in PSG data. This should at least be more adequately acknowledged as a limitation in the discussion. We do not think this invalidates the important findings described in the revised manuscript, but we do think this limitation should not be dismissed either.

Response: We followed the Reviewer's recommendation added explicit acknowledgement to the discussion that the lack of (i) *pulse oximetry or plethysmography measurements* and (ii) *expert evaluation by a trained clinician represent a clear limitation of our study, as we cannot definitively exclude any influence of SDB on our results (see page 14, line 327).*

*... It has to be noted that we did not explicitly screen for sleep disordered breathing (SDB) conditions in our participants, which might have considerably impacted the results of our work. However, SDB is well known to cause sleep fragmentation (e.g.,⁶⁹). All our participants exhibited healthy sleep and no signs of sleep fragmentation (i.e., elevated number of awakenings) as assessed with sleep scoring. Moreover, we carefully inspected the respiratory data during recording and offline pre-processing and found no signs of breathing cessation. Finally, sleep oscillations and specifically sleep spindle properties (i.e., frequency and topography) have been shown to be altered by SDB (e.g.,⁶⁸). Spindle characteristics were generally within the expected range, both in terms of frequency and topography. **That said, the fact that pulse oximetry or plethysmography were not collected during measurements, while no trained clinician evaluated the data, constitutes a limitation of the current study, as we cannot exclude any influence of SDB on our results with certainty ...***

Minor comments:

1. Some of the revised text includes typographical errors. One example is the caption under supplementary fig 5, which should read π to π and not $-p$ to π and the parentheses should be closed out.

Response: We thank the Reviewer and corrected the errors.

2. We think that SO_Spindle as a term in printed text is not typical of standard formatting. However, we think this is a decision best decided by the editors and the authors and not us as reviewers.

Response: We agree that this is not standard formatting but think it best conveys the nature of coupling between SOs and spindles.

REVIEWERS' COMMENTS

Reviewer #1 (Remarks to the Author):

I want to thank the authors again for their careful revision and gladly recommend publication.

Reviewer #2 (Remarks to the Author):

In my view, the authors have satisfactorily addressed all of my remaining points, and the ms is ready for publication.

Reviewer #3 (Remarks to the Author):

The authors have adequately addressed all of our concerns and we now endorse this important article for publication.